# BMP and FGF signaling interact to pattern mesoderm by controlling basic helix-loop-helix transcription factor activity

Richard H Row[1†], Amy Pegg[2†], Brian A Kinney[1], Gist H Farr 3rd[3], Lisa Maves[3,4], Sally Lowell[2*], Valerie Wilson[2*], Benjamin L Martin[1*]

[1]Department of Biochemistry and Cell Biology, Stony Brook University, Stony Brook, United States; [2]MRC Center for Regenerative Medicine, Institute for Stem Cell Research, School of Biological Sciences, University of Edinburgh, Edinburgh, United Kingdom; [3]Center for Developmental Biology and Regenerative Medicine, Seattle Children's Research Institute, Seattle, United States; [4]Division of Cardiology, Department of Pediatrics, University of Washington, Seattle, United States

**Abstract** The mesodermal germ layer is patterned into mediolateral subtypes by signaling factors including BMP and FGF. How these pathways are integrated to induce specific mediolateral cell fates is not well understood. We used mesoderm derived from post-gastrulation neuromesodermal progenitors (NMPs), which undergo a binary mediolateral patterning decision, as a simplified model to understand how FGF acts together with BMP to impart mediolateral fate. Using zebrafish and mouse NMPs, we identify an evolutionarily conserved mechanism of BMP and FGF-mediated mediolateral mesodermal patterning that occurs through modulation of basic helix-loop-helix (bHLH) transcription factor activity. BMP imparts lateral fate through induction of Id helix loop helix (HLH) proteins, which antagonize bHLH transcription factors, induced by FGF signaling, that specify medial fate. We extend our analysis of zebrafish development to show that bHLH activity is responsible for the mediolateral patterning of the entire mesodermal germ layer.
DOI: https://doi.org/10.7554/eLife.31018.001

**\*For correspondence:**
sally.lowell@ed.ac.uk (SL);
v.wilson@ed.ac.uk (VW);
benjamin.martin@stonybrook.edu (BLM)

[†]These authors contributed equally to this work

**Competing interests:** The authors declare that no competing interests exist.

## Introduction

The mesodermal germ layer gives rise to a host of adult tissues and organs that constitute the musculoskeletal, cardiovascular, and genitourinary systems, among others. Immediately after mesoderm induction begins during vertebrate gastrulation, the germ layer is patterned by secreted morphogenetic signals that promote different mediolateral fates. BMP acts as the major lateralizing signal to induce fates such as blood, vasculature, and pronephros (kidney), while FGF and canonical Wnt signaling induce more medial fates such as the notochord and somites (*Dorey and Amaya, 2010*; *Hikasa and Sokol, 2013*; *Tuazon and Mullins, 2015*). Despite advances in determining how the BMP and FGF signaling gradients are established, the molecular mechanisms directing mediolateral pattern formation in the mesoderm remain unknown.

Understanding patterning during gastrulation downstream of FGF and BMP signaling is complicated, since these pathways also affect the patterning of the anterior-posterior (AP) axis (*De Robertis, 2008*; *Kimelman, 2006*). The pleiotropic patterning roles during gastrulation make it difficult to interpret their effects on specific mesodermal fate decisions at later stages of development. Interaction between the pathways further confounds a simple readout of their effects. For example, FGF signaling represses transcriptional activation of BMP ligands, thereby inhibiting BMP

signaling (*Fürthauer et al., 2004*). FGF can also inhibit BMP signaling through MAPK activation, which phosphorylates the linker region of the BMP transducer SMAD, causing it to be targeted for proteasome and degraded (*Pera et al., 2003*).

We used a simplified model of mesodermal patterning to understand how FGF and BMP signaling induce mediolateral mesodermal fate. After gastrulation ends in vertebrate embryos, a structure called the tailbud forms at the posterior-most end of the embryo (*Beck, 2015*). The tailbud contains neuromesodermal progenitors (NMPs) that continue to make a germ layer decision between neural ectoderm and mesoderm during axis elongation ([*Martin and Kimelman, 2012*; *Tzouanacou et al., 2009*] and see (*Henrique et al., 2015*; *Kimelman, 2016*; *Martin, 2016*) for reviews). In zebrafish, NMP-derived mesoderm in the tailbud makes a further binary decision between lateral and paraxial fates, which is determined by canonical Wnt signaling levels (*Martin and Kimelman, 2012*). High Wnt signaling promotes the formation of paraxial mesoderm, which later gives rise to somites, while low Wnt signaling is required for adoption of lateral mesoderm fate, specifically endothelia (*Martin and Kimelman, 2012*). This binary decision, made when the AP axis is already established, allows us to focus specifically on mediolateral mesoderm patterning.

In this study, we show that that FGF and BMP signaling function in both zebrafish and mouse NMP-derived mesoderm to specify mediolateral pattern, with BMP promoting lateralization and FGF-inducing medial fate. Using NMPs, we also found that FGF induces medial fate through transcriptional activation of the bHLH transcription factors *myf5*, *myod*, and *msgn1*, and BMP counters this and promotes lateral fate through transcriptional activation of *Id* genes. *Id* genes encode HLH proteins that bind to and inhibit the function of bHLH transcription factors (*Ling et al., 2014*). We present a model based on our data of a conserved vertebrate mesodermal mediolateral patterning mechanism downstream of FGF and BMP that is based on the regulation of bHLH transcription factor activity.

## Results

### BMP signaling is necessary and sufficient for endothelial specification from NMP-derived mesoderm in zebrafish

We first examined the activity of BMP signaling, which induces lateral mesoderm during gastrulation (*Tuazon and Mullins, 2015*). To determine whether BMP signaling acts similarly in post-gastrula stage embryos, we examined mesodermal fate in embryos where BMP signaling was manipulated post-gastrulation using either a heat-shock inducible dominant negative BMP receptor transgenic line (*HS:dnbmpr*) or a small molecule inhibitor of BMP receptors (DMH1) (*Hao et al., 2010*; *Pyati et al., 2005*). Inhibition of BMP signaling in whole embryos using the heat-shock inducible transgenic line at the 12-somite stage resulted in a loss of endothelial tissue (including dorsal aorta and caudal vein), and in its place ectopic somite tissue formed (*Figure 1A–B*). The loss of blood vessels prevented the normal circulation of blood in the posterior body (*Figure 1—video 2* compared to *Figure 1—video 1*). Inhibition of BMP signaling using the small molecule inhibitor DMH1 phenocopied the dominant negative BMP receptor (*Figure 1C–L*). The DHM1-treated embryos contained transgenes to label skeletal muscle and endothelium, and revealed that ectopic somite tissue differentiates into skeletal muscle at the midline, where endothelium normally forms (*Figure 1J–L*, compared to E-G). The ectopic somite tissue is the same length along the AP axis as the bilateral somites that normally form, and the muscle reporter shows that the ectopic muscle tissue connects pairs of bilateral somites at the midline below the notochord. Loss of BMP signaling using either the *HS:dnbmpr* line or DMH1 at the end of gastrulation (bud stage) in embryos with an endothelial reporter transgenic background produced a more severe effect than the 12-somite stage heat-shock, with the gain of somite tissue and loss of endothelium occurring more anteriorly and across a broader domain of the AP axis (*Figure 1—figure supplement 1A–D*).

To provide further evidence that BMP signaling influences an NMP-derived mesodermal fate decision, we determined whether activation of BMP signaling is sufficient to specify vascular endothelium in NMP derived mesoderm. We created a new heat-shock inducible transgenic line (*hsp70: caAlk6-p2a-NLSkikume*), based on a previous transgenic line (*Row and Kimelman, 2009*), that can cell-autonomously activate BMP signaling. Heat-shock induction at the 12-somite stage caused a broad expansion of the early endothelial marker *etv2* 6 hr after induction, specifically in the region

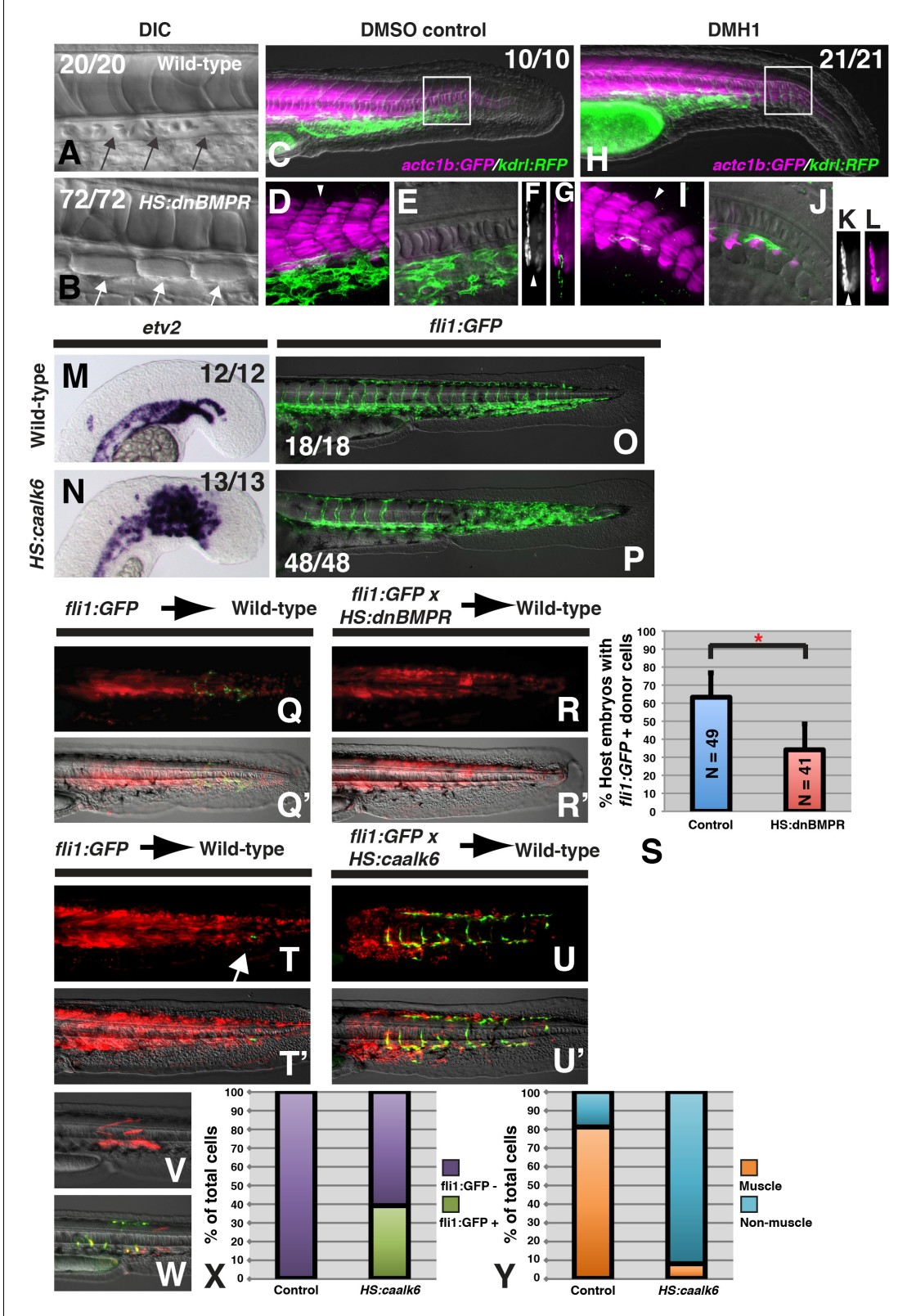

**Figure 1.** BMP signaling is necessary and sufficient for endothelial fate specification in tailbud-derived mesoderm. (**A**) Wild-type sibling embryos heat-shocked at the 12-somite stage exhibit normal formation of the dorsal aorta (black arrows, 20/20 normal). (**B**) *HS:dnbmpr* embryos heat-shocked at the 12-somite stage have ectopic segmented somite tissue where the dorsal aorta normally forms (white arrows, *72/72* with ectopic somite tissue). (**C–L**) Loss of BMP signaling using the small molecule DMH1 phenocopies *HS:dnbmpr* embryos. Embryos transgenic for both the *actc1b:GFP* (muscle,

*Figure 1 continued on next page*

*Figure 1 continued*

magenta) and *kdrl:rfp* (endothelium, green) transgenes were treated with DMSO (**C–G**) or DMH1 (**H–L**). A confocal Z-projection of the boxed region in C shows the presence of both muscle and endothelium in control DMSO treatment. A single z-slice at the midline shows the presence of endothelium and absence of muscle, which can also be observed in a digital cross section at the level of the white arrowhead in panel D. A confocal z-projection of the boxed region in H shows the presence of muscle and large reduction in endothelium (**I**). A single z-section at the midline shows the reduction of endothelium is accompanied by ectopic midline muscle formation, also observed in the digital cross-section at the level of the white arrowhead in panel I. (**M, N**) Transgenic *HS:caalk6* embryos heat-shocked at the 12-somite stage exhibit expansion of the endothelial marker *etv2* into the pre-somitic mesoderm 5 hr after the heat-shock (control N = 12, *HS:caalk6* N = 13). (**O, P**) At 36 hpf, *HS:caalk6* embryos heat-shocked at 12-somite stage have a dramatic expansion of *fli1:GFP* expression in posterior regions that would normally form somites, whereas there is no effect on anterior somites that formed before the heat-shock (Control N = 18, *HS:caalk6* N = 48). (**Q–R′**) Rhodamine dextran (red) labeled *fli1:GFP* donor cells were transplanted into unlabeled wild-type host embryos to monitor for contribution of transplanted cells to endothelium. (**Q, Q′, S**) Control cells contribute to endothelium in 63% of host embryos (N = 49). (**R, R′, S**) Heat-shock induction of *dnbmpr* at the 12-somite stage significantly (p=0.0107) reduces the percentage of host embryos (34%) that have donor-derived endothelium (N = 41). (**T–U′**) Induction of endothelium by BMP signaling is cell-autonomous, as exhibited in *HS:caalk6* x *fli1:GFP* cells transplanted wild-type host embryos. Host embryos were heat-shocked at the 12-somite stage and assayed for *fli1:GFP* expression at 36 hpf. (**U, U′**) *HS:caalk6* transgenic cells do not contribute to somites and instead give rise to endothelium. One-cell transplants were done to quantify fate changes after BMP activation (**W**) compared to controls (**V**). 12-somite stage BMP activation resulted in 39% *fli:GFP* positive cells (four embryos, 49 cells), compared to 0% in control transplants (three embryos, 36 cells, p<0.0001) (**X**). The fate of control transplanted cells was 81% muscle, whereas only 8% of *HS:caalk6* cells adopted a muscle fate (p<0.0001) (**Y**). All embryos are pictured from a lateral view with the head to the left, except for F, G, K, and L which are digital transverse sections.

DOI: https://doi.org/10.7554/eLife.31018.002

The following video and figure supplement are available for figure 1:

**Figure supplement 1.** BMP signaling inhibition at bud stage medializes tail mesoderm.
DOI: https://doi.org/10.7554/eLife.31018.003

**Figure 1—video 1.** Blood flow in a 48 hpf control embryo that was heat-shocked at the 12-somite stage.
DOI: https://doi.org/10.7554/eLife.31018.004

**Figure 1—video 2.** A movie illustrating the complete lack of posterior blood flow in a 48 hpf *HS:dnBMPR* transgenic embryo that was heat-shocked at the 12-somite stage.
DOI: https://doi.org/10.7554/eLife.31018.005

**Figure 1—video 3.** Blood flow in a 48 hpf wild-type embryos heat shocked at the 12-somite stage.
DOI: https://doi.org/10.7554/eLife.31018.006

**Figure 1—video 4.** Blood flow in a 48 hpf *HS:caalk6* transgenic embryo heat-shocked at the 12-somite stage.
DOI: https://doi.org/10.7554/eLife.31018.007

**Figure 1—video 5.** High-magnification view of blood flow in the tail of a 48 hpf wild-type embryo that was heat-shocked at the 12-somite stage.
DOI: https://doi.org/10.7554/eLife.31018.008

**Figure 1—video 6.** High-magnification view of blood flow in the tail of a 48 hpf *HS:caalk6* transgenic embryo heat-shocked at the 12-somite stage, focusing on a position where the skeletal muscle of somites would normally exist in a wild-type embryo.
DOI: https://doi.org/10.7554/eLife.31018.009

of the pre-somitic mesoderm but not in already formed somites (*Figure 1M,N*). Performing the same 12-somite heat-shock in a *fli1:GFP* background revealed that at 36 hr post-fertilization (hpf), the paraxial mesoderm is converted to vascular endothelium in regions posterior to the 12[th] somite (*Figure 1O,P*). The expanded vascular endothelium forms a massive network of functional blood vessels, as observed by imaging blood flow in live embryos (*Figure 1—videos 4* and *6* compared to *Figure 1—videos 3* and *5*).

To determine whether BMP signaling acts cell-autonomously during endothelial specification of NMP derived mesoderm, we transplanted cells from sphere stage donor embryos to the ventral margin of shield stage host embryos. This manipulation targets cells to the future tailbud NMP population (*Martin and Kimelman, 2012*). A homozygous *fli1:GFP* line was crossed to a hemizygous *HS:dnBMPR* or hemizygous *HS:caalk6* line and embryos from these crosses were injected with rhodamine dextran. The injected embryos were used as donors and transplanted into unlabeled wild-type host embryos. This method allows us to visualize all of the transplanted cells (red fluorescence) as well as any transplanted cells that adopt an endothelial fate (green fluorescence). Host embryos were heat-shocked at the 12-somite stage and imaged at 36 hpf. Loss of BMP signaling significantly reduced the number of host embryos that contained donor derived endothelial tissue (*Figure 1Q–S*). Conversely, activation of BMP signaling at the 12-somite stage caused a large number of transplanted cells to become endothelium, and prevented donor cells from integrating into somites and

forming skeletal muscle (*Figure 1T–U′*). These results show that BMP signaling plays a cell-autonomous role (without ruling out additional non-autonomous roles) in endothelial induction.

To quantify the extent of fate change, we used a previously reported one-cell transplant method to create small clones of cells that could be accurately counted after BMP signaling activation (*Martin and Kimelman, 2012*). Single cells from *fli1:GFP* embryos or *fli1:GFP* crossed to *HS:caalk6* embryos were transplanted into the ventral margin of shield stage wild-type host embryos, which were heat-shocked at the 12-somite stage. Cells were analyzed for muscle cell morphology or *fli1: GFP* labeling at 36 hpf (*Figure 1V,W*). Control cells gave rise to 81% muscle and 0% endothelial fate, whereas cells with activated BMP signaling were 8% muscle and 39% endothelium (*Figure 1X, Y*). The remaining cells in each condition were of mixed non-muscle, non-endothelial fates, which based on their position and morphology appear to be primarily fin mesenchyme. Together, the results from loss- and gain-of-function experiments indicate that BMP signaling is necessary and sufficient to induce endothelium from NMP-derived mesoderm, and that paraxial mesoderm maintains plasticity to become endothelium in response to BMP signaling at least until somite formation occurs.

## Mouse NMP-derived mesoderm exhibits plasticity and is lateralized by BMP signaling

As in zebrafish, mesoderm generated from mouse NMPs is almost exclusively paraxial in character (*Cambray and Wilson, 2007*; *Tzouanacou et al., 2009*). To test whether mouse NMPs can differentiate to lateral fates, we microdissected NMPs from embryos carrying a ubiquitous GFP marker and grafted them in posterior primitive streak regions fated to become lateral mesoderm (*Gilchrist et al., 2003*). The grafted cells incorporated readily into lateral mesoderm, showing that these cells, like zebrafish NMPs, can adopt lateral as well as more medial fates (*Figure 2A*). To determine whether FGF and BMP signaling can influence mouse NMP-derived mesodermal patterning, we made use of NMPs derived in vitro from pluripotent cells (*Gouti et al., 2014*). In vitro-derived NMPs were treated with FGF2 alone, or FGF2 and BMP4. BMP4 promoted expression of nascent mesoderm markers at the expense of neural markers. This suggests that BMP4 blocks neural differentiation in NMPs, as it does in the gastrulation stage epiblast (*Di-Gregorio et al., 2007*).

We then examined markers of mesoderm subtypes. Compared to FGF treatment alone, FGF plus BMP-treated NMPs showed higher expression of lateral mesodermal markers *Flk1* (expressed in endothelia) and *Hand1* (expressed widely in the lateral mesoderm). This treatment also resulted in lower levels of the paraxial mesoderm markers *Meox1* and *Tcf15* (*Figure 2B*). In NMPs, as expected, SOX2 was co-expressed with BRACHYURY (T) but after differentiation in FGF, these markers segregated into distinct populations of neural (SOX2) and nascent mesodermal (T) cells. FGF-treated cells predominantly expressed the paraxial mesoderm marker MEOX1, with few cells expressing FLK1 (*Figure 2E*). In contrast, cells exposed to BMP lost both SOX2 and T during differentiation (*Figure 2C*) and did not upregulate MEOX1 (*Figure 2E*), but instead expressed FLK1. Using a *Flk1-GFP* reporter ES cell line (*Jakobsson et al., 2010*) we confirmed that the majority of the BMP-treated cells differentiated into *Flk1* +lateral mesoderm (*Figure 2D,E*). Together, these results imply that BMP-mediated lateralization of NMP-derived mesoderm is a conserved vertebrate phenomenon.

## FGF signaling prevents zebrafish presomitic mesoderm from adopting an endothelial fate

The induction of paraxial mesoderm fate from NMPs in vitro by FGF signaling alone suggested that this factor may be important for paraxial mesoderm patterning. Indeed, during gastrulation, FGF signaling promotes paraxial mesoderm fates (*Fürthauer et al., 1997*; *2004*; *Lee et al., 2011*), and is active in the post-gastrulation paraxial presomitic mesoderm (*Dubrulle et al., 2001*; *Sawada et al., 2001*). Interestingly, during posterior axial extension, BMP ligands are expressed in ventral posterior tissues juxtaposed to the presomitic mesoderm (*Martínez-Barberá et al., 1997*). The activation of BMP signaling in the presomitic mesoderm indicated that this tissue can be induced to form endothelium when subjected to high levels of BMP signaling. This suggested that the presomitic mesoderm may require protection from the adjacent lateralizing signal by another molecular factor. To determine whether FGF acts in this way, we inhibited FGF signaling during zebrafish axial extension

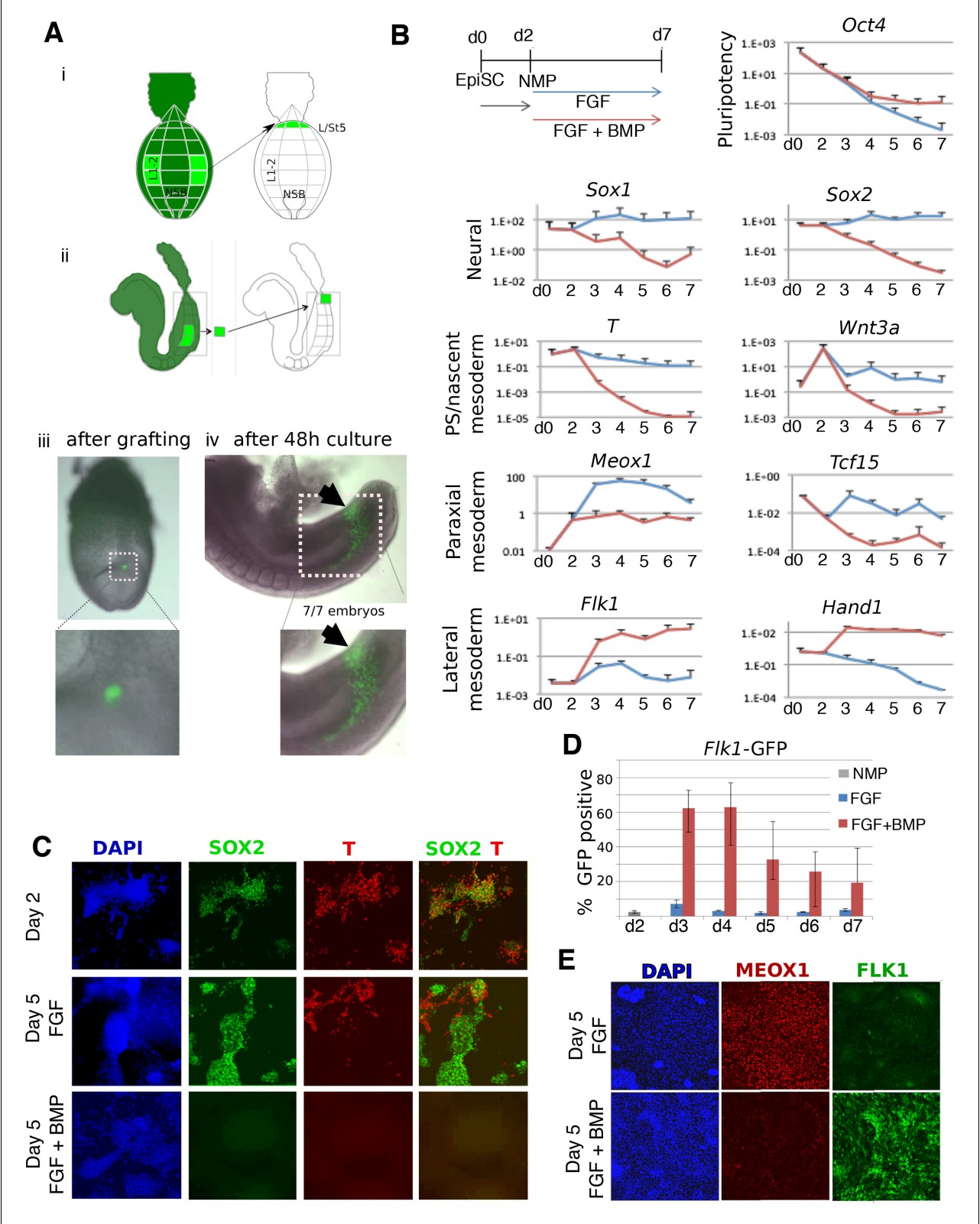

**Figure 2.** BMP redirects fate of mouse NMPs from paraxial to lateral mesoderm. (**A**) Heterotopic grafting from ubiquitous GFP embryos (*Gilchrist et al., 2003*) of NMP region fated for paraxial mesoderm at 2–5 somite stage (E8.0) into the posterior primitive streak region fated for lateral and ventral mesoderm, followed by 48 hr culture. (i) Posterior view (ii) Lateral view (iii) Representative embryo immediately after grafting showing position of GFP +grafted cells (iv) Representative embryo after 48 culture showing that descendants of grafted cells have adopted a lateral fate
*Figure 2 continued on next page*

Figure 2 continued

(arrowheads). (B) qPCR at indicated time points during the differentiation of EpiSCs into NMPs then treated with FGF2 or FGF2 +BMP4. Data shown relative to the housekeeping gene TBP. (C) Immunofluorescence detection of indicated markers in in vitro derived NMPs and their differentiating derivatives. (D) Flow cytometry of *Flk1-GFP* in the differentiating derivatives of in-vitro derived NMPs. (E) Immunofluorescence detection of indicated markers in the differentiating derivatives of in-vitro derived NMPs.

DOI: https://doi.org/10.7554/eLife.31018.010

The following source data is available for figure 2:

**Source data 1.** Raw data for the qPCR experiments in *Figure 2*.
DOI: https://doi.org/10.7554/eLife.31018.011

using a heat-shock inducible dominant negative FGF receptor transgenic line (*Lee et al., 2005*). Transgene expression was induced at the 12-somite stage and embryos were fixed 6 hr later and examined for *etv2* expression. The loss of FGF signaling phenocopied the gain of BMP signaling with a strong expansion of the endothelial marker *etv2* into the PSM (*Figure 3A,B*). The same result is achieved with a small molecule FGF receptor inhibitor (not shown) or MEK inhibitor (*Figure 3C,D*), indicating that specification of tailbud-derived mesoderm to paraxial identity by FGF occurs through the MAPK pathway.

To determine whether FGF signaling plays a cell-autonomous role in the medialization of tailbud-derived mesoderm, we performed transplants as with the BMP transgenic lines. Embryos from a *HS: dnfgfr* to *fli1:GFP* cross or *HS:dnfgfr* to *kdrl:gfp* cross were used as donors for transplantation into the ventral margin of unlabeled wild-type host embryos. Heat-shock induction of the *dnfgfr* transgene at the 12-somite stage caused a significant cell-autonomous shift from somite to endothelial fate (*Figure 3E–I*). Together these results indicate that FGF signaling plays a cell autonomous role (without ruling out additional non-autonomous roles) in maintaining paraxial fate through MAPK signaling in plastic presomitic cells.

## Lateral mesoderm is the default state of zebrafish tailbud-derived mesoderm

As previously mentioned, FGF signaling can inhibit BMP signaling during gastrulation (*Fürthauer et al., 2004*; *Pera et al., 2003*). To test whether FGF signaling maintains paraxial fate through BMP signaling inhibition, we first examined phospho-SMAD (pSMAD) 1,5,8 staining in MEK inhibitor treated embryos. Unexpectedly, there is no expansion of pSMAD 1,5,8 staining after loss of MEK activity, and it appears down-regulated in the posterior-most regions of the embryo (*Figure 3J,K*). Consistent with a lack of expansion of BMP signaling, expression of *id1*, a member of the inhibitor of DNA binding (*id*)1–4 family of BMP target genes, appears down-regulated within the posterior mesoderm, although interestingly *id1* expression appears expanded in the prospective neural forming region of the tailbud (*Figure 3—figure supplement 1*).

To confirm that expansion of *etv2* expression after the loss of FGF signaling is not due to an increase or expansion of BMP signaling, we simultaneously inhibited both FGF and BMP signaling. Embryos at the 12-somite stage were treated with either the MEK inhibitor, DMH1, or both and assayed 6 hr later for *etv2* expression. The expansion of *etv2* into the PSM after MEK inhibition was not blocked by the BMP inhibitor DMH1 (*Figure 3O* compared to N). We also tested the combination of MEK inhibitor and the BMP and VEGF inhibitor dorsomorphin. To ensure that dorsomorphin was inhibiting BMP signaling (as it inhibits both BMP and VEGF signaling), embryos were pretreated with dorsomorphin starting at bud stage. The loss of pSMAD 1/5/8 in the tailbud was confirmed at the 12-somite stage (*Figure 3—figure supplement 2A,B*, arrow). At the 12-somite stage, a MEK inhibitor was added to the embryos with dorsomorphin and they were grown for an additional 6 hr before fixation and examination of *etv2* expression. Again, inhibition of BMP signaling failed to block MEK inhibitor mediated *etv2* expansion into the PSM (*Figure 3—figure supplement 2F* compared to E). These results suggest that the expansion of vasculature after the loss of FGF signaling is not due to a secondary increase in BMP signaling, and that in the absence of FGF signaling, BMP signaling is dispensable for endothelial induction. Thus, with respect to BMP and FGF signaling, lateral mesoderm is the default fate of tailbud-derived mesoderm.

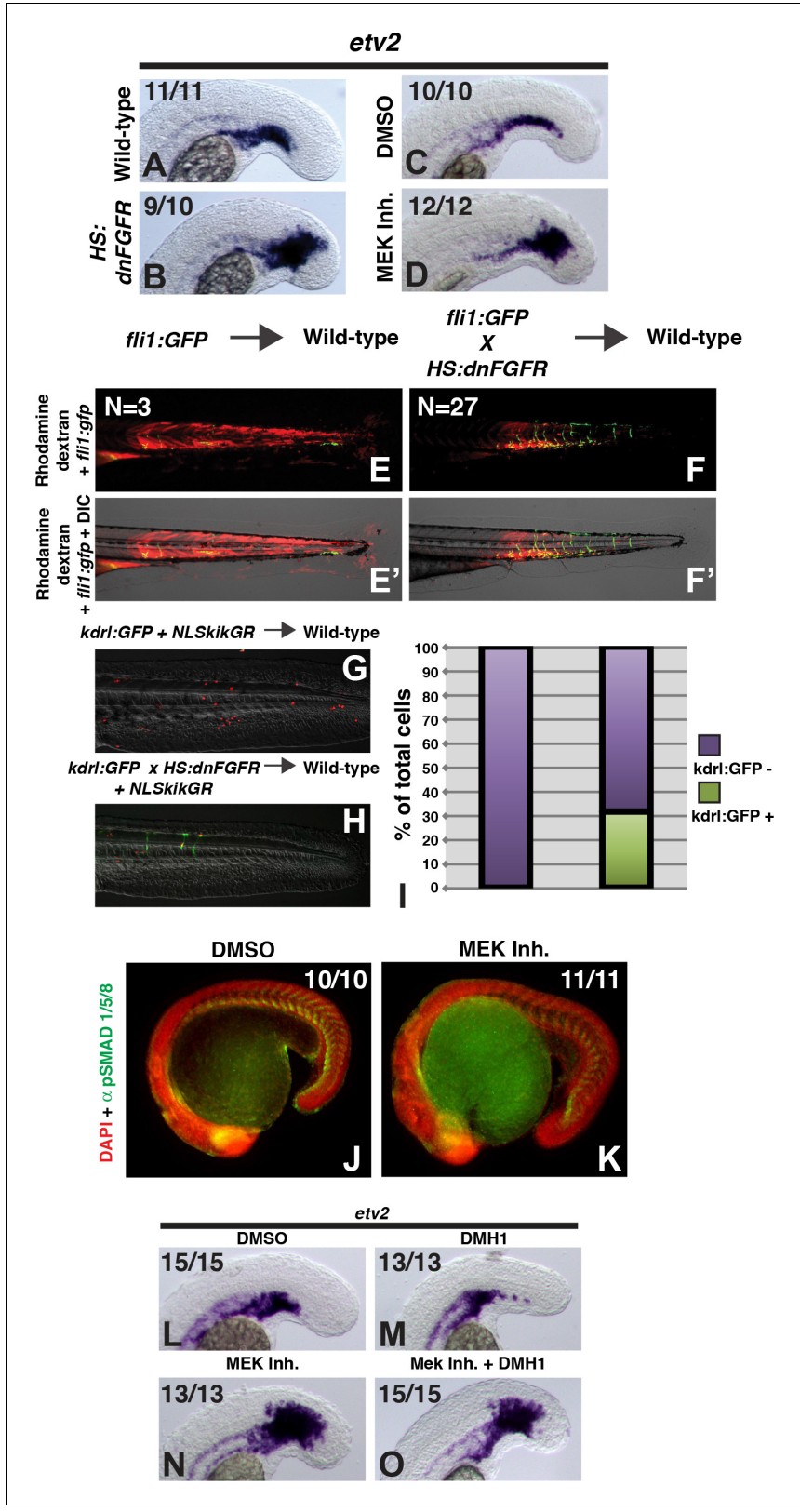

**Figure 3.** FGF signaling is necessary to maintain paraxial mesoderm fate and inhibit a default endothelial fate. Heat-shock induction of *dnfgfr* (B) or treatment with a MEK inhibitor (D) at the 12-somite stage causes an expansion of the endothelial marker *etv2* into the pre-somitic mesoderm 5 hr later compared to controls (A, C). (F, F') Transplanted *HS:dnfgfr* x *fli1:GFP* show a cell-autonomous shift from somite to endothelial fate when heat-

*Figure 3 continued on next page*

*Figure 3 continued*

shocked at the 12-somite stage, whereas *fli:GFP* transplants mostly contribute to muscle with minor endothelial contribution (**E, E'**). The same effect is seen with *HS:dnfgfr* x *kdrl:GFP* transplanted cells when heat-shocked at the 12-somite stage (**G, H**). *NLS-kikume* was injected into donor embryos to quantify cell fate changes. 12-somite stage FGF inhibition resulted in 31% *kdrl:GFP*-positive cells (13 embryos, 308 cells), compared to 0% in control transplants (seven embryos, 587 cells, p<0.0001) (**I**). Expansion of endothelium 5 hr after MEK inhibitor treatment is not due to an expansion of BMP signaling, as revealed by pSMAD 1/5/8 staining (**K** compared to **J**, green staining, red color is DAPI staining). (**L–O**) Similarly, treatment with the BMP inhibitor DMH1 does not prevent MEK-inhibitor-induced expansion of endothelium. Embryos were treated at the 12-somite stage and fixed 6 hr later. The expansion of the endothelial marker *etv2* into the PSM after MEK inhibitor treatment (**N**) is not inhibited by the addition of DMH1 (**O**).

DOI: https://doi.org/10.7554/eLife.31018.012

The following figure supplements are available for figure 3:

**Figure supplement 1.** MEK inhibitor does not cause an expansion of *id1* expression into the PSM.
DOI: https://doi.org/10.7554/eLife.31018.013

**Figure supplement 2.** Dorsomorphin does not rescue MEK-inhibitor-induced endothelial expansion.
DOI: https://doi.org/10.7554/eLife.31018.014

## BMP signaling lateralizes mesoderm through activation of *id* gene transcription in both mouse and zebrafish

Our result that endothelium is the default state of tailbud-derived mesoderm suggests that BMP is not directly activating an endothelial specific program, but rather acting as an inhibitor of FGF-induced medial (presomitic) mesoderm. We considered known direct BMP target genes that could have a negative effect on both the Wnt and FGF pathways, or their downstream target genes. Amongst these, *id1, 2, 3* and *4* are candidates. ID genes encode helix-loop-helix (HLH) proteins that act as endogenous dominant negative inhibitors of basic helix-loop-helix (bHLH) transcription factors (*Ling et al., 2014*; *Wang and Baker, 2015*). ID proteins bind to and inhibit E proteins, which are ubiquitously expressed bHLH transcription factors that are essential dimerization partners for tissue specific bHLH transcription factors (*Norton, 2000*). FGF and Wnt signaling activate many paraxial-specific bHLH transcription factors such as *msgn1*, *myoD*, and *myf5*, among others (*Dietrich et al., 1998*; *Fior et al., 2012*; *Geetha-Loganathan et al., 2005*; *Hoppler et al., 1996*; *Marcelle et al., 1997*; *Marics et al., 2002*; *Mok et al., 2014*; *Münsterberg et al., 1995*; *Pan et al., 2015*; *Steinbach et al., 1998*; *Tajbakhsh et al., 1998*; *Wittler et al., 2007*). The most prominently expressed ID genes in the zebrafish tailbud are *id1* and *id3* (*Thisse et al., 2001*; *Thisse and Thisse, 2004*). Using heat-shock inducible transgenic lines, we showed that BMP signaling is necessary and sufficient for both *id1* and *id3* expression in the tailbud (*Figure 4—figure supplement 1*).

To directly test whether *id* activation downstream of BMP signaling mediates the paraxial/endothelial fate decision, we designed an assay based on transient transgenic embryos and cell transplantation. We made heat-shock inducible plasmids using the *hsp70l* promoter to drive an *id1-p2a-NLS-kikume* or *id3-p2a-NLS-kikume* construct flanked by *tol2* transposable element arms. The single *id1-p2a-NLS-kikume* transcript produces two independent proteins, Id1 and NLS-Kikume, based on the cleavable viral peptide p2a (the same is true for the *id3* construct). We injected each plasmid and *tol2* transposase mRNA into *fli1:GFP* embryos to create transiently transgenic embryos and transplanted cells from these embryos into the ventral margin of wild-type host embryos. Host embryos were heat-shocked at the 12-somite stage and analyzed at 36 hpf. The NLS-Kikume was photoconverted from green to red. The nuclear localization of the photoconverted Kikume was used to quantify the number of transplanted transiently transgenic cells. Cells with red nuclei that also contained cytoplasmic GFP indicated those transplanted cells that adopted an endothelial fate. Empty *hsp70l:p2a-NLS-kikume* plasmid was injected and transient transgenic cells were transplanted as a control. Tailbud-derived mesoderm normally gives rise mostly to paraxial mesoderm and only a small percentage of endothelium (*Figure 4A,A', G*) (*Martin and Kimelman, 2012*). Activation of either Id1 or Id3 at the 12-somite stage resulted in a drastic increase in the percentage of cells becoming endothelium (*Figure 4C,C', E,E', G*). When BMP signaling is simultaneously inhibited using the *HS:dnBMPR* transgenic line at the same time as Id1 or Id3 is activated, Id activation is still able to induce a similar percentage of cells to contribute to endothelium (*Figure 4D,D', F,F', G*). These results

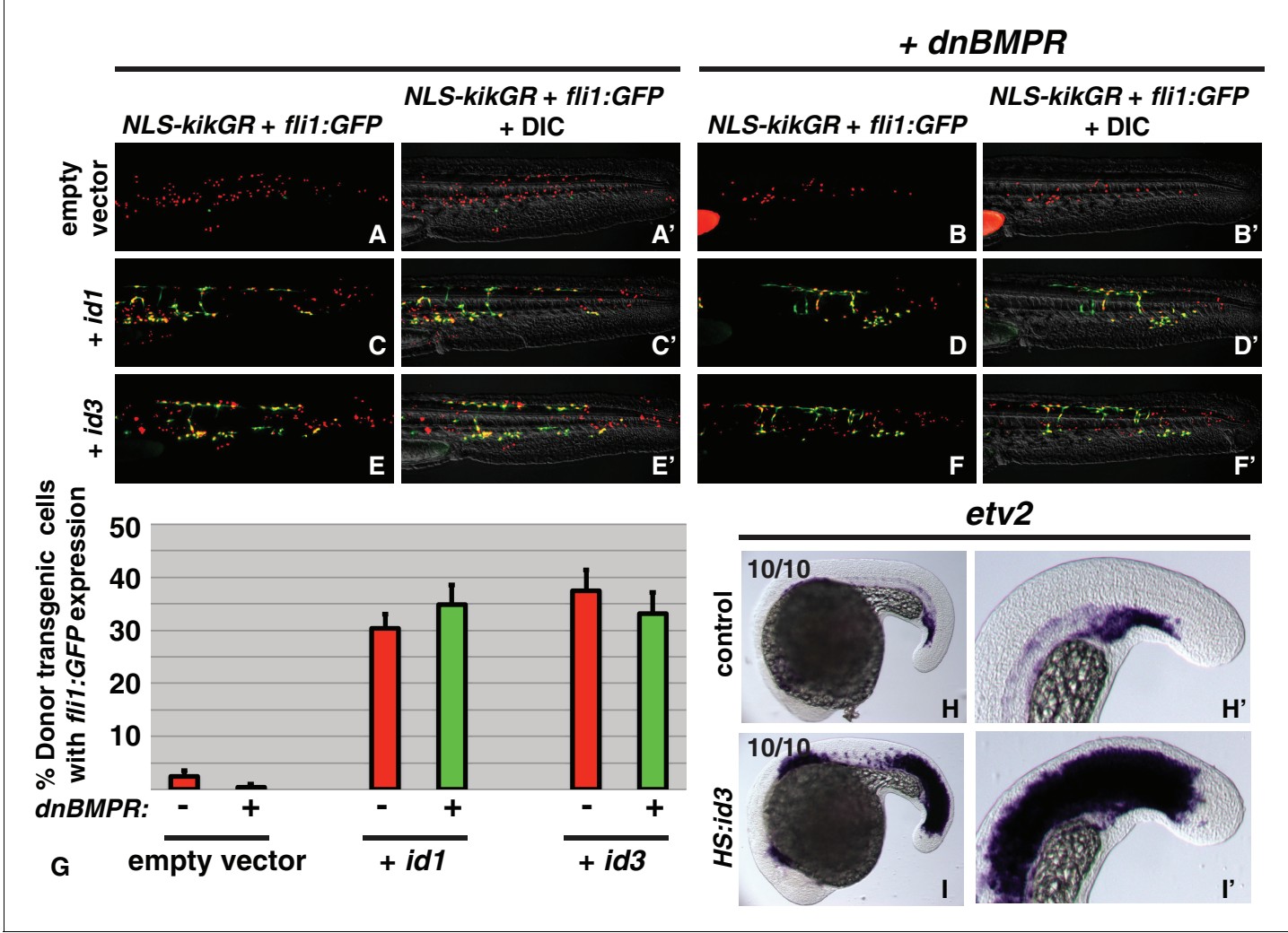

**Figure 4.** *id* genes are the essential BMP targets mediating endothelial induction. An assay was developed to quantify the percent of transplanted cells that adopt an endothelial fate (see text for details). Control cells transplanted to the ventral margin of host embryos and heat-shocked at the 12-somite stage exhibit a small percentage contribution to endothelium (green cells, **A, A', G**, empty vector $N^{embryos}$ = 19, $N^{cells}$ = 500), which is significantly reduced when BMP signaling is inhibited in transplanted cells (**B, B', G**, empty vector +*dnbmp*r $N^{embryos}$=19, $N^{cells}$ = 1022, p=0.006). Activation of *id1* or *id3* causes a significantly larger percentage of transplanted cells to adopt an endothelial fate (**C, C', E, E', G**, *id1* $N^{embryos}$ = 16, $N^{cells}$ = 1159, p<0.0001, *id3* $N^{embryos}$ = 18, $N^{cells}$ = 574, p<0.0001), and this effect is unchanged in cells that also lack BMP signaling (**D, D', F, F', G**, *id1* +*dnbmpr* $N^{embryos}$=16, $N^{cells}$ = 614, p<0.0001, *id3* +*dnbmpr* $N^{embryos}$=12, $N^{cells}$ = 531, p<0.0001). Cell fate quantification from these experiments is represented in panel G. A stable *HS:id3* transgenic line heat-shocked at the 12-somite stage and fixed 5 hr later exhibits a large expansion of the endothelial marker *etv2* (**I, I'**) compared to heat-shocked wild-type embryos (**H, H'**).

DOI: https://doi.org/10.7554/eLife.31018.015

The following figure supplement is available for figure 4:

**Figure supplement 1.** BMP signaling is necessary and sufficient for *id1* and *id3* expression.

DOI: https://doi.org/10.7554/eLife.31018.016

indicate that Id proteins are the critical cell-autonomous factors inducing endothelium downstream of BMP signaling. To further examine the role of Id proteins during mesodermal patterning, we generated a stable *hsp70l:id3-p2a-NLS-kikume* transgenic line. Heat-shock induction at the 12-somite stage resulted in a dramatic expansion of *etv2* expression 6 hr after the heat-shock in both the presomitic and somitic domains (***Figure 4H–I'***).

Since mouse NMP-derived mesoderm is also lateralized by BMP signaling (***Figure 2***), we wanted to decipher whether the same downstream BMP target genes mediate lateralization. Similar to zebrafish, mouse *Id1* and *Id3* are expressed in the most posterior (lateral mesoderm fated) region of

the primitive streak. They were not detected in anterior primitive streak in regions fated to become paraxial mesoderm (*Figure 5A*). We then made use of in vitro derived NMPs to determine whether BMP activates expression of *Id1* in mouse NMPs (*Figure 5B*) (*Gouti et al., 2014*). *Id1* transcript levels (*Figure 5C*) and activity of an *Id1-Venus* reporter (*Malaguti et al., 2013*) (*Figure 5D*) were both increased in response to BMP. The *Id1-Venus* reporter revealed, however, that *Id1* could be detected in a subset of cells even in the absence of exogenous BMP (*Figure 5D,E*), at least partly due to low levels of endogenous BMP in the culture (data not shown). We made use of this heterogeneity in *Id1* expression to ask whether suppression of paraxial mesoderm differentiation occurs only in *Id1*-expressing cells. Indeed, we observed that the paraxial mesoderm marker MEOX1 was restricted to the *Id1*-negative cells within FGF-treated cultures. (*Figure 5E*). These results show that BMP signaling induces *Id1* expression in mouse NMPs, and that induction of *Id1* correlates with suppression of paraxial mesoderm markers in individual cells.

To directly assess whether ID1 mediates the effect of BMP to lateralize mouse NMP derived mesoderm, we used cells lines engineered for doxycycline inducible over-expression of a flag-tagged *Id1* transgene (*Malaguti et al., 2013*). We first confirmed that addition of dox was able to induce *Id1* to levels comparable to those induced by BMP during differentiation of NMPs in the presence of FGF (*Figure 6A–C*: compare with *Figure 5D*). Activation of *Id1* expression was able to recapitulate the effect of exogenous BMP on differentiation: cells expressing the *Id1* transgene in the absence of exogenous BMP robustly increased expression of endothelial marker *Flk1* and largely lacked the paraxial mesoderm marker *Meox1* (*Figure 6D,E*). The mouse and zebrafish data show that BMP mediates lateralization of NMP-derived mesoderm via activation of Id expression and suggests that this mechanism acts across vertebrates.

## Zebrafish FGF signaling medializes mesoderm through transcriptional activation of bHLH transcription factors

Although Id proteins are best known for their ability to bind to and inhibit bHLH transcription factors, they can also bind to other proteins including Retinoblastoma (*Ruzinova and Benezra, 2003*). To determine whether BMP regulated *id* expression lateralizes mesoderm through bHLH transcription factor inhibition and not an alternative mechanism, we wanted to identify bHLH transcription factors regulated by FGF signaling required for its medializing activity. We examined the expression of bHLH transcription factors *msgn1*, *myf5*, and *myod* 6 hr after a 12-somite stage treatment with a MEK inhibitor. *msgn1* expression is completely abolished, and there is a near complete loss of *myf5* expression in MEK inhibited embryos (*Figure 7A,B,D,E*). The expression of *myod* is only moderately reduced in MEK inhibited embryos (*Figure 7F* compared to C). In both mouse and zebrafish, loss of function of either *myf5* or *myod* has relatively mild skeletal muscle phenotypes, whereas loss of both genes results in a complete absence of skeletal muscle (*Braun et al., 1992*; *Hinits et al., 2011*; *Maves et al., 2007*; *Rudnicki et al., 1992*; *Rudnicki et al., 1993*). We examined an endothelial marker (*kdrl*) in *myf5/myod* double homozygous mutants and found a significant expansion of endothelium into normal skeletal muscle territories (*Figure 7G–J*), although the expansion was not as prominent as that induced by activation of BMP signaling or Id3 expression (see *Figure 1* and *Figure 4*).

To determine whether *msgn1* functions in a partially redundant manner with *myf5* and *myod* to medialize NMP derived mesoderm, we used previously characterized translation blocking antisense morpholinos (MOs) to disrupt combinations of these three gene products (*Maves et al., 2007*; *Yabe and Takada, 2012*). As previously reported in both morphants and mutants, loss of either *myod* or *myf5* alone did not cause a significant loss of muscle (*Hinits et al., 2011*; *Maves et al., 2007*). Similarly, loss of *msgn1* alone produced only a mild enlarged tailbud phenotype, as previously reported (*Fior et al., 2012*; *Yabe and Takada, 2012*). The expression of *etv2* was examined at the 22-somite stage. Loss of *msgn1/myod* resulted in a mild expansion of *etv2* (*Figure 7L,L'*). The expansion was broader in *myf5/myod* loss of function embryos (*Figure 7M,M'*). In embryos lacking *msgn1/myf5*, which were the two most affected genes after MEK loss of function (*Figure 7A–E*), there is broad expansion of *etv2* throughout the normal PSM and somite region (*Figure 7N,N'*). Loss of all three bHLH genes enhanced the expansion of *etv2* and resulted in a phenocopy of heat-shock induction of *id3* (*Figure 7O,O'*). To determine whether there is a corresponding loss of muscle in these embryos, we stained them for *ttna* expression, which labels skeletal muscle, and observed a reduction in expression that was strongest in the triple knockdown (*Figure 7—figure supplement*

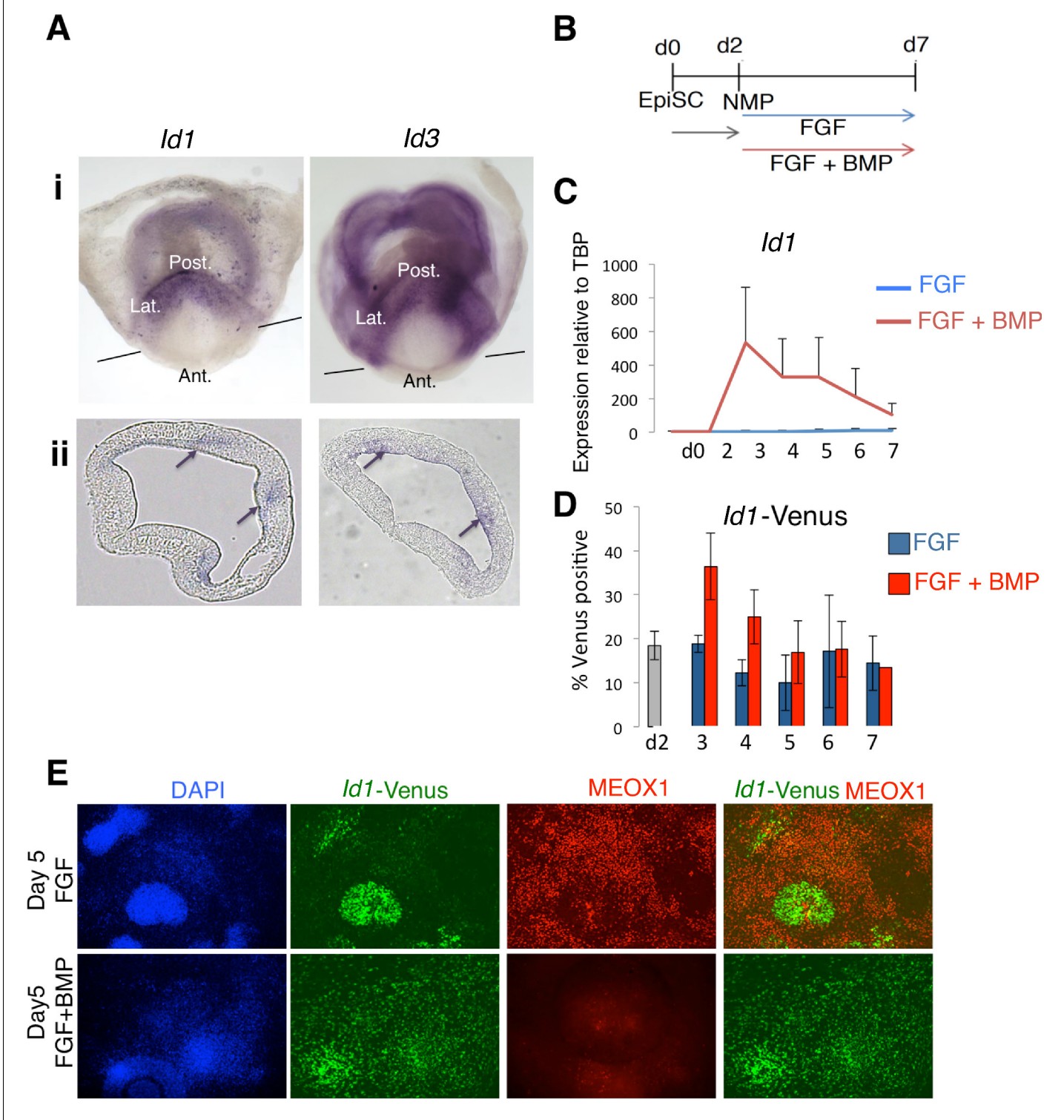

**Figure 5.** *Id1* and *Id3* are expressed in prospective lateral/ventral mesoderm at early somite stages in vivo, and are induced by BMP in NMPs in vitro. (A) In situ hybridisation for *Id1* and *Id3* in wholemount (i) and sections (ii) showing *Id1* and *Id3* expression restricted to the posterior (labelled 'Post.') and lateral (labeled 'Lat.') regions of the primitive streak. *Id1/3* are not detected in the anterior primitive streak (labeled 'Ant'). Lines in (i) indicate the plane of section. Arrows in (ii) indicate regions of expression in the posterior lateral regions of the primitive streak. (B) In vitro differentiation protocol. (C) *Id1* mRNA is expressed in response to BMP4 but not FGF2 during differentiation of NMP in culture (D) an *Id1-Venus* reporter is activated in response to BMP4 but not FGF2 during differentiation of NMP in culture (E) Immunofluorescence for indicated markers during differentiation of NMP in culture: expression of MEOX1 is mutually exclusive from expression of ID1-Venus, and is suppressed by addition of BMP4.

*Figure 5 continued on next page*

*Figure 5 continued*

DOI: https://doi.org/10.7554/eLife.31018.017

The following source data is available for figure 5:

**Source data 1.** Raw data for the qPCR experiments in *Figure 5*.

DOI: https://doi.org/10.7554/eLife.31018.018

1). The same combinations of MOs were injected in *kdrl:GFP* transgenic fish and stained for skeletal muscle (MF20 antibody) at 36 hpf. These embryos revealed that there is a large expansion of differentiated endothelium at the expense of differentiated skeletal muscle (*Figure 7P–U*). Additionally, the lateralized phenotype of *HS:id3* embryos can be rescued by co-activation of *msgn1* (*Figure 7— figure supplement 2*).

To determine whether *myf5/msgn1/myod* function cell-autonomously in the medialization of mesoderm, we performed transplants with morpholino injected donor cells. Either *myod* MO alone or all three MOs were co-injected along with rhodamine dextran into *kdrl:gfp* embryos, and cells from these embryos were transplanted into the prospective mesoderm of wild-type host embryos. Cells lacking *myod* function behave normally and primarily join somites and form skeletal muscle, with a small number of cells contributing to endothelium (*Figure 7V*, endothelium is green). On the other hand, cells lacking all three gene products are completely excluded from the somites and populate the majority of the vasculature throughout the length of the embryo (*Figure 7W*), and these blood vessels appear to be functional (*Figure 7—video 1*). These results indicate that *msgn1*, *myf5*, and *myod* play a partially redundant, cell-autonomous role in the specification of somitic mesoderm and inhibition of lateral endothelial mesoderm. bHLH transcription factor activity functions broadly within the zebrafish mesodermal germ-layer to pattern the mediolateral axis

In order to determine whether bHLH activity acts broadly within the mesodermal germ layer, we examined the expression of the red blood cell marker *gata1a*. Post-gastrula NMP derived mesoderm never gives rise to primitive red blood cells (*Martin and Kimelman, 2012*). Double *myod;myf5* mutants stained for skeletal muscle (*actc1b*) and red blood cells (*gata1a*) at 24 hpf show a complete loss of muscle and a gain of red blood cell staining, whereas single mutants and wild-type embryos have normal muscle and red blood cells (*Figure 8A–D*). Analysis of *myod/myf5/msgn* triple-MO injected embryos at the 22-somite stage show expanded *gata1a* in the vicinity of the normal expression domain, as in *myod;myf5* double mutants, as well as ectopic expression in the normal muscle forming domains (*Figure 8F,F'* compared to E, E'). Finally, *myod/myf5/msgn* triple MOs were injected into *gata1a:DsRed;kdrl:gfp* double transgenic embryos along with cascade blue dextran and used as donors in a chimeric analysis. Cascade blue injected control donor cells targeted to the mesodermal territory contribute primarily to skeletal muscle, with minor contributions to endothelium and red blood cells (*Figure 8G*, endothelium in green (green arrow) and red blood cells in red (red arrow)). On the other hand, *myod/myf5/msgn* triple-MO donor cells are excluded from skeletal muscle and contribute extensively to endothelium and red blood cells of host embryos (*Figure 8H*). These results indicate that the bHLH transcription factors Myod, Myf5, and Msgn1 are normally required to promote medial (somitic) fate and inhibit lateral fates such as endothelium and primitive blood.

Based on loss of bHLH transcription factor analysis, we expected activation of *id3* expression during gastrulation to also lateralize mesoderm in a manner consistent with activation of BMP signaling during gastrulation, where BMP activation inhibits notochord and somite formation and expands pronephros, endothelial, and hematopoietic tissues (*Dosch et al., 1997*; *Neave et al., 1997*). Activation of *id3* expression at shield stage in cells transplanted into wild-type host embryos resulted in an absence of cells in the trunk musculature and extensive contribution to the vasculature (*Figure 8J* compared to I). We next examined a panel of markers at 24 hpf representing a spectrum of mediolateral mesoderm types in whole *HS:id3* transgenic embryos that were heat-shocked at the start of gastrulation (shield stage). There is a loss of the medial mesoderm tissues of notochord (*ntla*) and muscle (*myod*), and a gain in lateral mesoderm tissues of pronephros (*pax2a*), endothelium (*kdrl*), and red blood cells (*gata1a*) (*Figure 8K–T'*). The expansion of the red blood cell marker was most significant, encompassing the majority of the somitic territory in the trunk of the embryo (*Figure 8T'* compared to S'). These results indicate that bHLH transcription factor activity is involved in

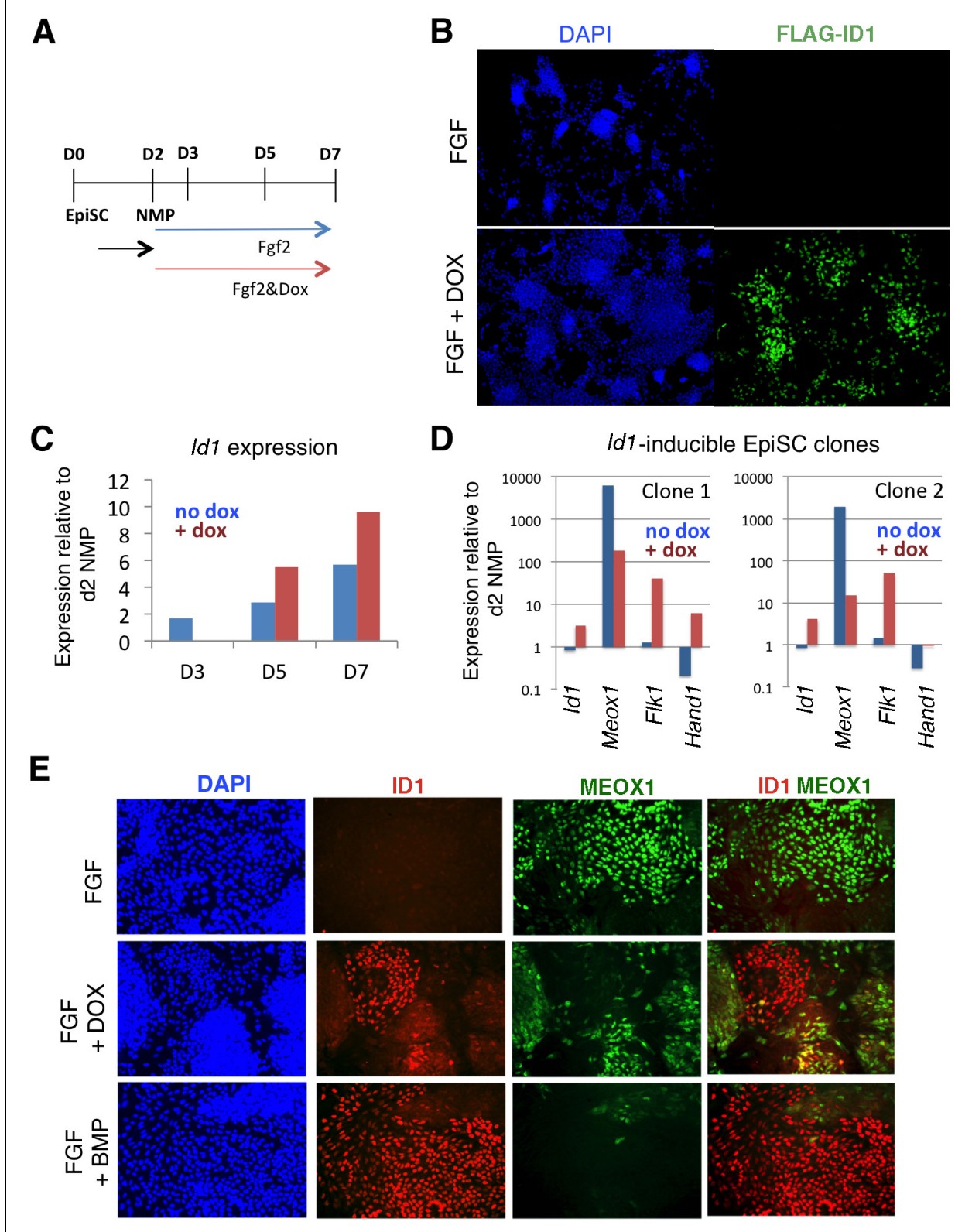

**Figure 6.** *Id1* GOF drives differentiation of lateral mesoderm at the expense of paraxial mesoderm. A: Differentiation protocol. B: Immunofluorescence detection for the Flag epitope in *Flag-Id1* inducible EpiSC indicates that addition of dox induces Flag-ID1 in a subset of cells. C: qPCR to detect *Id1* mRNA in the absence and presence of dox in *Flag-Id1* inducible EpiSC. D: qPCR to detect the indicated mesoderm markers in *Flag-Id1* inducible

*Figure 6 continued on next page*

*Figure 6 continued*

EpiSC the absence and presence of dox: data from two independent clonal lines is shown. D: Immunofluorescence detection of indicated markers: induction of *Id1* suppresses expression of MEOX1, recapitulating the effect of adding BMP4.

DOI: https://doi.org/10.7554/eLife.31018.019

The following source data is available for figure 6:

**Source data 1.** Raw data for the qPCR experiments in *Figure 6*.

DOI: https://doi.org/10.7554/eLife.31018.020

mediolateral patterning throughout the entire mesodermal germ layer, and that there are likely other bHLH transcription factors in addition to *myf5*, *myod*, and *msgn1* that are involved in mesoderm patterning.

Previous work demonstrated that BMP signaling patterns the mediolateral axis in tight coordination with anterior-posterior axis development, such that BMP provides mediolateral pattern progressively from the head to the tail over time as the body develops (*Hashiguchi and Mullins, 2013*; *Tucker et al., 2008*). Our data indicate that *id* genes are the critical BMP targets that mediate its role in mediolateral pattern, and thus activation of *id3* should also lateralize mesoderm progressively from the head to the tail. We ubiquitously activated *id3* expression using the *HS:id3* transgenic line at progressively later stages of development (*Figure 8—figure supplement 1*). Activation during gastrulation, when the anterior-most mesoderm is being specified, caused a loss of muscle and expansion of vasculature anteriorly, but patterning was relatively normal in posterior domains. At the end of gastrulation, activation of *id3* resulted in loss of muscle and gain of endothelium throughout the axis except for the most anterior and posterior regions. Finally, activation of *id3* during post-gastrula stages, when the posterior mesoderm is being specified, resulted in posterior loss of muscle and expansion of endothelium, but normal patterning anteriorly. To demonstrate that the recovery of posterior patterning after gastrula stage *id3* activation was due to the turnover of the transgene, we performed two heat-shock inductions (during and after gastrulation), which prevented recovery of patterning in posterior mesoderm. These results, combined with our epistasis analysis in *Figure 4*, indicate that *id* genes are the critical BMP targets that account for the activity of the BMP pathway in mediating mediolateral mesodermal patterning. The BMP-induced Id proteins promote lateral fate by antagonizing the FGF-induced bHLH transcription factors, which promote medial fate (*Figure 8U*).

## Discussion

### *Id* genes are the essential targets mediating BMP induced lateralization of mesoderm

ID proteins, as inhibitors of bHLH transcription factors, are best known for prolonging the progenitor state of lineage committed cells. For example, ID1 inhibits myogenesis in myoblasts by inhibiting the activity of MYOD (*Benezra et al., 1990a*; *Benezra et al., 1990b*). ID proteins also inhibit neurogenesis and prolong the neuroblast state through the inactivation of several different neurogenic bHLH factors (*Bai et al., 2007*; *Jung et al., 2010*; *Liu and Harland, 2003*). Additionally, ID1 maintains the undifferentiated state of ES cells through antagonism of FGF induced TCF15 (*Davies et al., 2013*). These activities prevent precocious differentiation of stem cells or lineage specified progenitors and allow expansion of progenitor populations prior to differentiation. However, there are few examples of *Id* genes affecting cell fate decisions. The primary example is during the white blood cell lineage decision between NK and B cell fate, where mouse ID2 promotes NK-cell fate over B-cell fate through inhibition of E2A (*Boos et al., 2007*; *Ikawa et al., 2001*; *Yokota et al., 1999*), which itself promotes B-cell fate (*Zhuang et al., 1994*). The role of ID proteins in fate determination has likely been obscured by the fact that the four vertebrate *Id* genes play partially redundant roles, such that any single mouse mutant lacks a severe phenotype. Significant embryonic phenotypes can only be observed in multiple knockouts. The *Id1/Id3* double knockout mouse has angiogenesis defects in the brain (*Lyden et al., 1999*). The redundancy of ID proteins was also recently substantiated by transient CRISPR/Cas9-mediated loss of ID1-4 function during mouse embryogenesis, which causes defects in cardiac mesoderm specification (*Cunningham et al., 2017*).

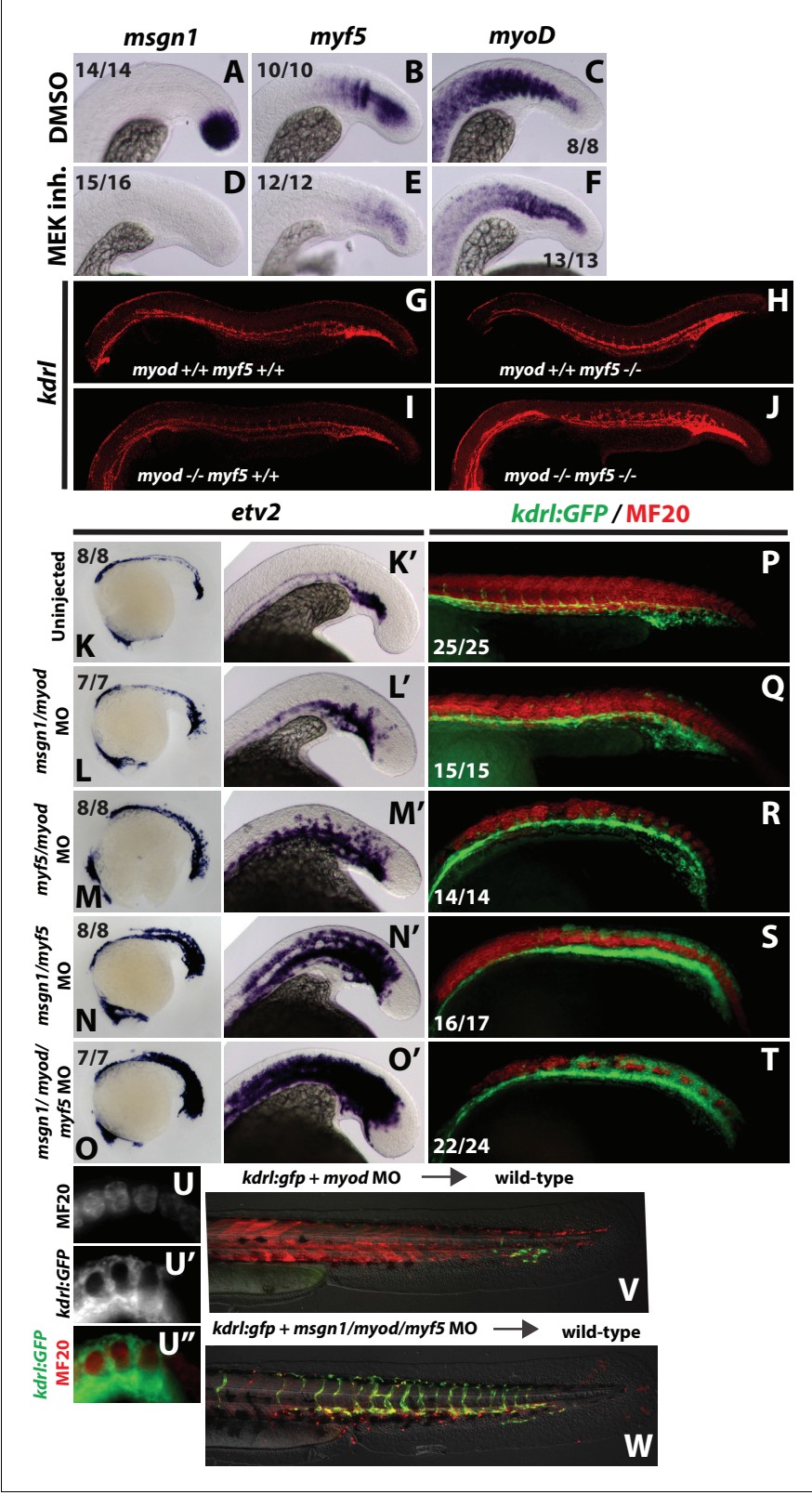

**Figure 7.** FGF signaling maintains paraxial mesoderm fate and inhibits endothelial fate through positive regulation of bHLH transcription factors. Wild-type embryos were treated with the MEK inhibitor or DMSO at the 12-somite stage and fixed five hours later. Expression of *msgn1* (D) and *myf5* (E) were significantly downregulated compared to controls (A, B), whereas *myod* (F) exhibited only a minor reduction in expression compared to controls (C). Expression of *kdrl* (red) is expanded into somitic territories in *myod;myf5* double mutants compared to controls (G). n = 44 controls (pooled +/+;+/
*Figure 7 continued on next page*

*Figure 7 continued*

+, +/+;+/-; +/-;+/+, and +/-;±genotyped embryos). 0/44 controls have expanded *kdrl*. n = 4 mutants (-/-;-/- genotyped embryos). 4/4 show expanded *kdrl* (representative embryos shown). MO-mediated loss of function of *msgn1/myod* (L, L') or *myf5/myod* (M, M') results in a moderate expansion of *etv2* expression at the 22 somite stage, whereas loss of *msgn1/myf5* causes a broad expansion of *etv2* (N, N'). Loss of function of all three genes further enhances *etv2* expansion (O, O'). MF20 (muscle, red) antibody staining in 30 hpf *kdrl:GFP* embryos demonstrates the gain of differentiated vasculature at the expense of differentiated muscle (P–T). U-U'' are high-magnification views of MF20 staining and *kdrl:GFP* expression in a *msgn1/myf5* loss of function embryo. Transplanted *kdrl:GFP* cells lacking *myod/myf5/msgn1* fail to join host somites and instead contribute predominantly to endothelium (W), whereas cells lacking myod behave normally, with most transplanted cells joining the somites and forming muscle (V).

DOI: https://doi.org/10.7554/eLife.31018.021

The following video and figure supplements are available for figure 7:

**Figure supplement 1.** bHLH transcription factor knockdown inhibits skeletal muscle specification.

DOI: https://doi.org/10.7554/eLife.31018.022

**Figure supplement 2.** Over-expression of *msgn1* rescues *id3* over-expression.

DOI: https://doi.org/10.7554/eLife.31018.023

**Figure 7—video 1.** Blood flow in a 30 hpf host embryos that received donor *kdrl:GFP* cells that were injected with *msgn1*, *myod*, and *myf5* MOs.

DOI: https://doi.org/10.7554/eLife.31018.024

The patterning role that Id proteins play in mesoderm lateralization is due at least in part to the inhibition of Myod, Myf5, and Msgn1 function. Based on mediolateral patterning phenotypes, our data suggest that Id inhibition of Myf5 is more important during lateralization than inhibition of Myod. Although Myod and Myf5 are redundantly required for skeletal myogenesis, recent work using mouse cells showed that they have distinct functions in regulating the muscle specific transcriptional program. During muscle differentiation, MYF5 initially modifies chromatin through histone acetylation, but does not act as a strong transcriptional activator. MYOD binds to the same genomic sites as MYF5 and can modify chromatin, but in the presence of MYF5 its primary role is to recruit POLII to strongly activate muscle-specific transcription (*Conerly et al., 2016*; *Gerber et al., 1997*). Thus, Id-mediated lateralization is likely achieved by preventing a paraxial mesoderm competent chromatin state through Myf5 inhibition. Since MSGN1 plays an essential role in mouse to promote a paraxial mesoderm specific transcriptional program essential for somite development (*Chalamalasetty et al., 2014*; *Yoon et al., 2000*; *Yoon and Wold, 2000*), Myf5 and Msgn1 likely function together to modify chromatin and create the paraxial competent state.

## Do ID proteins mediate the morphogen activity of BMP signaling?

BMP functions as a morphogen during mediolateral mesodermal patterning. Different levels of signaling have distinct outputs on cell fate (*Dale et al., 1992*; *Dale and Wardle, 1999*; *Dosch et al., 1997*; *James and Schultheiss, 2005*; *Neave et al., 1997*). For example, cells receiving the highest level of BMP signaling adopt a blood or endothelial fate, intermediate levels specify pronephros and paraxial tissues, and the absence of BMP signaling is required for notochord fate. Here, we show that *Id* genes are critical downstream targets of BMP that mediate its role in mediolateral patterning of the mesoderm. A key unresolved question is whether Id protein levels produce the morphogenetic output of BMP signaling during mediolateral mesodermal patterning. Based on the ability of BMP signaling to act as a morphogen, we envision two plausible scenarios. Id proteins mediate the morphogenetic activity of BMP signaling within the mesoderm, with different levels of Id protein specifying distinct mesodermal cell fates. An alternative scenario is that Id proteins levels do not act alone as the mediators of morphogenetic output, but rather impact a series of binary fate decisions that are dictated by the position of the cell in the mesodermal territory. In this case, the morphogenetic activity of BMP signaling integrates both its role in cell fate and cell migration, where the level of BMP signaling dictates the position of a cell in the embryo and mediates the binary decision that is possible in that precise location/signaling environment. BMP signaling has been previously shown to be capable of controlling cell migration during zebrafish gastrulation independent of cell fate (*von der Hardt et al., 2007*). This important question can be resolved in the future through the controlled modulation of Id protein levels and the examination of cell fate output.

If Id proteins levels modulate the morphogenetic activity of BMP within the mesoderm, the mechanism will not be as simple as Id antagonism of the same set of bHLH transcription factor function throughout the entire mesodermal germ layer. We show that Msgn1, Myf5, and Myod act as

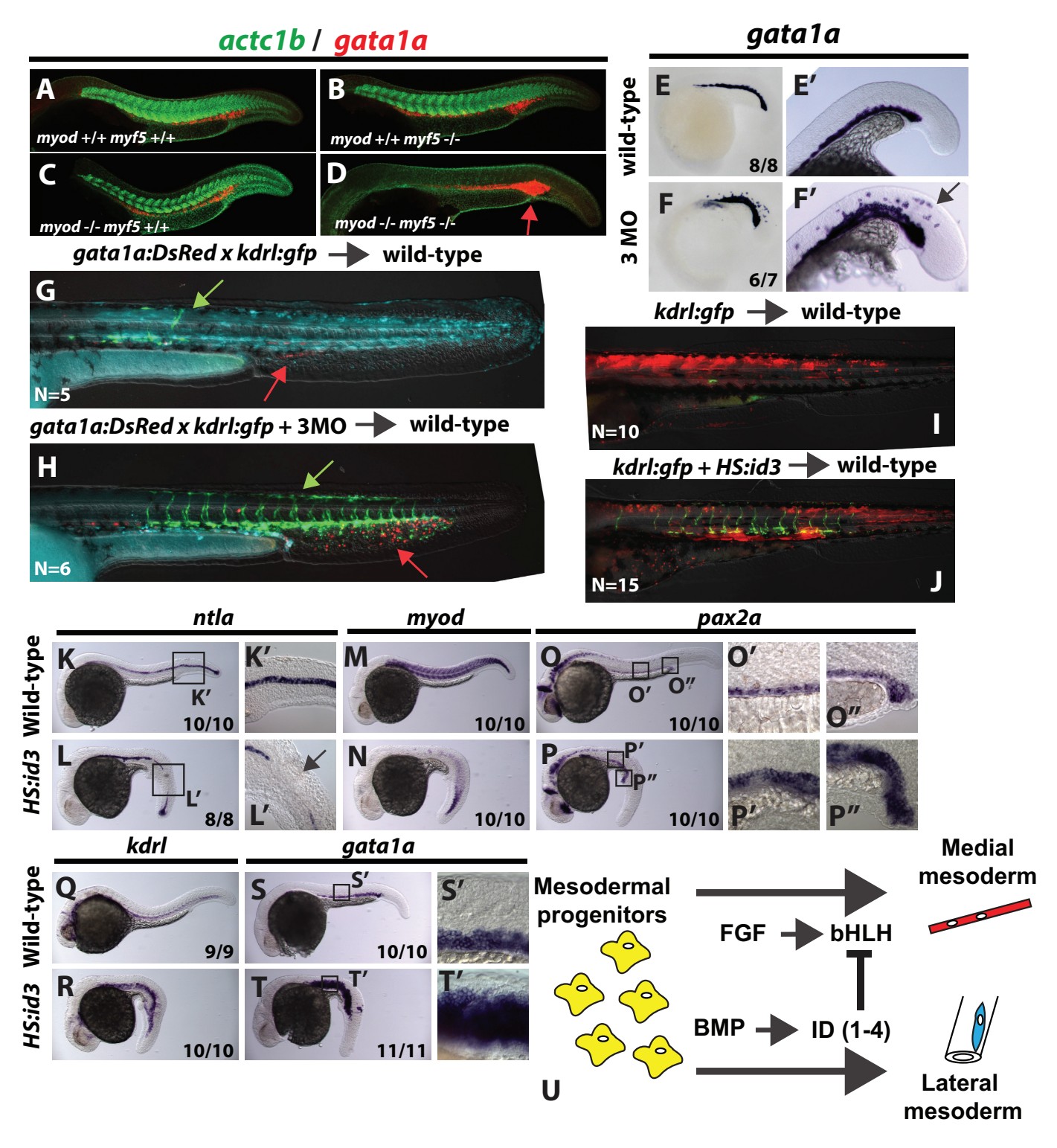

**Figure 8.** bHLH transcription factor activity provides mediolateral pattern to the entire mesodermal germ layer. Homozygous *myod;myf5* mutant embryos exhibit slightly expanded *gata1a* expression (D, red staining, arrow) and a complete loss of skeletal muscle marker *actc1b* (green staining). n = 49 controls (pooled +/+;+/+, +/+;+/-; +/-;+/+, and +/-;±genotyped embryos). 49/49 controls show normal *actc1b*, 0/49 controls have expanded *gata1a*. n = 9 mutants (-/-;-/- genotyped embryos). 9/9 show loss of *actc1b*, 7/9 show expanded *gata1a* (representative embryos shown). Loss of *msgn1/ myod/myf5* function results in an expansion of *gata1a* expression into somitic domains at the 22 s stage (E–F'). Cells from transgenic *gata1a:dsRed* x *kdrl:GFP* embryos injected with cascade blue dextran and *msgn1/myod/myf5* MOs transplanted into unlabeled host embryos are excluded from

*Figure 8 continued on next page*

*Figure 8 continued*

somites and contribute extensively to endothelium (H, green arrow) and red blood cell lineages (H, red arrow). Control cascade blue injected *gata1a: dsRed* x *kdrl:GFP* transplanted cells contribute primarily to somitic muscle with minor contributions to endothelium (G, green arrow) and red blood cells (G, red arrow). Heat-shock induction of *id3* at shield stage in mesodermally targeted transplanted cells that also contain the kdrl:GFP transgene causes a shift from predominantly somitic muscle fate to significant endothelial contribution in the trunk (J compared to I). Whole embryo induction of *id3* expression at shield stage and analyzed at 24 hpf indicates a loss of medial mesoderm (notochord and muscle, (K–N), and an expansion of lateral mesoderm (pronephros, vasculature, and blood, (O–T')). Expression of *gata1a* in the trunk shows broad expansion into somite territories (S' compared to T'). (U) A model for how FGF and BMP signaling control mediolateral patterning of the mesoderm through modulation of bHLH transcription factor activity.

DOI: https://doi.org/10.7554/eLife.31018.025

The following figure supplement is available for figure 8:

**Figure supplement 1.** *id3* mediated patterning of the mediolateral mesodermal axis is coordinated with AP axis formation.

DOI: https://doi.org/10.7554/eLife.31018.026

medializing factors in the mesoderm, but these transcription factors are not expressed in axial mesoderm. Given that Id3 activation inhibits notochord formation, there must be other bHLH transcription factor(s) in the axial mesoderm that promote the medial-most fate. Additionally, the lateralization caused by Id3 activation during gastrulation in non-axial mesoderm is more severe than loss of Msgn1, Myf5, and Myod function, indicating that additional bHLH gene(s) likely functions in a partially redundant fashion with Msgn1, Myf5, and Myod to impart paraxial/myogenic fate. On the other hand, loss of Msgn1, Myf5, and Myod fully phenocopies Id3 activation during postgastrulation development, indicating that these three genes are the primary determinants of somite fate within NMP-derived mesoderm.

## Evolutionarily conserved mediolateral patterning of NMP-derived mesoderm

Vertebrate NMP-derived mesoderm gives rise primarily to paraxial mesoderm, which forms the somites. We previously showed in zebrafish that a small percentage of NMP-derived mesoderm becomes endothelium, indicating a patterning event that occurs to generate medial paraxial and lateral endothelial mesoderm (*Martin and Kimelman, 2012*). Similarly, in mouse, a minority of mesodermal cells in NMP-derived clones contribute to lateral mesoderm (*Tzouanacou et al., 2009*). Here we show, using embryonic tissue transplantation and NMP cell culture, that mouse NMP-derived mesoderm is also plastic and can robustly differentiate to both medial and lateral mesodermal types, supporting previous grafting studies showing that lateral fated tailbud mesoderm can be re-specified to paraxial fate (*Wymeersch et al., 2016*). Furthermore, we show using in vitro mouse assays and in vivo zebrafish assays, that FGF signaling acts as a medializing factor, and BMP signaling as the lateralizing factor during mediolateral patterning. In both zebrafish and mouse, the lateralizing activity of BMP signaling can be phenocopied by over-expression of the BMP target gene *Id1*, indicating that bHLH transcription factor activity is a key conserved mechanism by which NMP-derived mesoderm is patterned along the mediolateral axis. Since NMPs have been observed in several other vertebrate species and are likely present in all vertebrates, we expect that the mediolateral fate decision in NMP-derived mesoderm to become paraxial mesoderm or posterior endothelium is common feature of vertebrate body formation.

We showed that in zebrafish, FGF signaling medializes and promotes somitic fate in NMP-derived mesoderm through transcriptional activation of the bHLH transcription factors *myf5*, *msgn1*, and *myod*. This indicates that in addition to its fundamental role in promoting cell motility and orchestrating segment formation during somitogenesis (*Bénazéraf et al., 2010*; *Dubrulle et al., 2001*; *Hubaud and Pourquié, 2014*), FGF signaling also maintains paraxial fate in unsegmented presomitic mesoderm. This function is solely dependent on regulation of bHLH transcription factor expression and not on inhibition of BMP signaling, which FGF signaling normally inhibits during gastrulation (*Fürthauer et al., 2004*; *Pera et al., 2003*). Loss of both FGF and BMP signaling lateralizes the NMP-derived mesoderm to the same extent as loss of FGF signaling alone. This suggests that with respect to FGF and BMP signaling, lateral mesoderm is the default fate. Why then is BMP signaling necessary in the tailbud to promote an endothelial fate? Both FGF and Wnt signaling, which are medializing factors, are also required for the induction of mesoderm from NMPs (*Goto et al., 2017*;

*Martin and Kimelman, 2012*), and therefore BMP signaling is required to counter the action of these signals after mesoderm induction. Indeed, loss of BMP signaling causes cells that would normally become endothelium to adopt a medial, somite fate.

## Methods

### Animal care

All zebrafish procedures were performed with prior approval from the Stony Brook University and Seattle Children's Research Institute Institutional Animal Care and Use Committee.

### Generation of zebrafish heat-shock inducible constructs and transgenic lines

We generated heat-shock inducible constructs as previously described (*Row et al., 2016*). Briefly, the coding sequence of zebrafish *id1*, *id3*, or a mutant constitutively active human *ALK6* (*caalk6*) without their stop codons were inserted into a heat-shock vector to create *hsp70l:id1-p2a-NLS-kikume*, *hsp70l:id3-p2a-NLS-kikume*, and *hsp70l:caalk6-p2a-NLS-kikume* (abbreviated as *HS:id1*, *HS:id3*, and *HS:caalk6*, respectively) flanked by *tol2* transposable element arms. Stable transgenic lines of *HS:id3* and *HS:caalk6* were generated by injecting the plasmid DNA along with in vitro transcribed *tol2* transposase mRNA (25 pg of each per embryo) into one-cell stage embryos and screening injected animals when they became adults for germ-line transmission (*Kawakami, 2004*).

### Zebrafish cell transplantation and statistics

In order to target cells to the future tailbud, cells from fluorescent dextran labeled (either Rhodamine, Fluorescein, or Cascade Blue dextran, MW 10,000, Molecular Probes) sphere stage donor embryos were transplanted into the ventral margin of unlabeled shield stage host embryos, as previously described (*Martin and Kimelman, 2012*). Transplantations were performed under a Leica S6E dissecting microscope using a Cell Tram Vario (Eppendorf). Statistical analysis of quantified cell transplants was performed with the Fisher's exact test.

### Zebrafish drug treatments

BMP signaling was inhibited using DMH1 (EMD Millipore) or dorsopmorhin (Chemdea LLC). A 1 mM stock solution of DMH1 was made in DMSO and diluted to a 10 uM working solution. A 5 mM stock solution od dorsomorphin was made in DMSO and diluted to a 10 uM working solution. FGF signaling was inhibited using PD173704 (LC Laboratories). A 10 mM stock in DMSO was diluted to a 100 uM working solution. The MAP Kinase cascade was inhibited using the MEK inhibitor PD325901 (LC Laboratories). A 10 mM stock solution in DMSO was diluted to a 25 uM working solution. For each drug treatment, controls were performed by treating embryos with an equivalent volume of DMSO in embryo media.

### Zebrafish lines, heat-shock conditions, and morpholinos

All wild-type embryos used in this study were from hybrid adults generated from an inbred strain of locally acquired pet store fish (which we call Brian) crossed to the TL line (TLB). Transgenic heat-shock inducible lines include *HS:dnfgfr*, *HS:dnbmpr*, *HS:TCFΔC*, *HS:caalk6*, and *HS:id3*. All heat shock inductions were performed at 40°C for 30 min, except for the *HS:dnfgfr* line which was heat-shocked at 38°C. To observe cell fate in live embryos, we used the *fli:GFP*, *kdrl:GFP* and *kdrl:RFP* transgenic reporter lines to monitor for endothelial fate, the *gata1a:dsRed* line to observe red blood cells, and the *actc1b:GFP* to visualize skeletal muscle.

The *myod^{fh261}* and *myf5^{hu2022}* mutant strains were maintained on the AB background and were previously described (*Hinits et al., 2009*; *Hinits et al., 2011*). *myod^{fh261}* genotyping was performed using forward primer 5'AACCAGAGGCTGCCCAAAGTGGAGATTCGG' and reverse primer 5'CCATGCCATCAGAGCAGTTGGATCTCG3'. The genotyping PCR product is 166 base pairs, and digesting with HphI yields a 136 base pair product from the mutant allele. *myf5^{hu2002}* genotyping was performed using forward primer 5'GCACTTGCGCTTCGTCTCC3' and reverse primer 5'CATCGGCAGGCTGTAGTAGTTCTC3'. When digested with BstAPI, the mutant allele PCR product is 365 base pairs and the wild-type allele products are 229 and 136 base pairs.

The *myod, myf5,* and *msgn1* morpholinos were validated and used as previously described (*Hinits et al., 2009*; *Maves et al., 2007*; *Yabe and Takada, 2012*).

## Zebrafish in situ hybridization and immunohistochemistry

Colorimetric in situ hybridization was performed as previously described (*Griffin et al., 1995*). Fluorescent whole-mount in situ hybridization was performed as previously described (*Talbot et al., 2010*). Following staining of *myod;myf5* mutant embryos, tissue from embryos was lysed and genotyped for *myod* and *myf5* as above. The following cDNA probes were used: *actc1b*; *kdrl* (*Thompson et al., 1998*); and *gata1a* (PCR probe provided by Scott Houghtaling).

Embryos for immunohistochemistry were fixed overnight in 4% paraformaldehyde at 4°C and stored in 100% methanol at −20°C. Embryos were rehydrated stepwise in PBS-tween and blocked for 1 hr at room temperature. Embryos were incubated in a 1:50 dilution of MF20 (Developmental Studies Hybridoma Bank – a myosin heavy chain antibody labeling skeletal and cardiac muscle) or anti-phospho SMAD 1/5/8 (Cell Signaling Technology, Inc.) at a 1:200 dilution overnight at 4°C. Primary antibodies were detected with Alexa-fluor conjugated secondary antibodies (Molecular Probes) used at a 1:500 dilution and incubated overnight at 4°C.

## Mouse cell lines

Id1 inducible cells are described in (*Malaguti et al., 2013*). Doxycyline was used at 1 μg/ml in order to induce expression of a flag tagged Id1 transgene during differentiation. Flk1-GFP cells were a gift from Alexander Medvinsky (*Jakobsson et al., 2010*), and Id1-Venus cells were generated as described in (*Malaguti et al., 2013*) using a targeting construct described in (*Nam and Benezra, 2009*).

## Mouse epiblast stem cell culture

Plates were coated with bovine fibronectin (Sigma) in PBS (7.5 μg/ml) at 37°C for at least 10 min prior to use. Epiblast stem cells were maintained in pre-prepared six well plates in serum-free media containing 20 μg/ml Activin A (Peprotech) and 10 μg/ml Fgf2 (R and D) (*Gouti et al., 2014*). EpiSC were passaged using Accutase (Sigma). Where counting of EpiSCs was required, a small aliquot of the fragmented re-suspension was transferred into fresh N2B27 and dissociated to single cells. Cell counting was performed using a haemocytometer (Neubauer).

## Mouse neuromesodermal progenitor differentiation

Differentiation of EpiSCs into neuromesodermal progenitors (NMPs) was carried as described in (*Gouti et al., 2014*). Briefly, EpiSC were plated on fibronectin in N2B27 medium supplemented with 20 ng/ml Fgf2 (R and D) and 3 μM Chiron (CHIR99021) (Axon Medchem). Cells were dissociated, pelleted and resuspended into fragmented clumps before plating at 1500 cells/cm$^2$. Cells were then cultured for 48 hr, at which point co-expression of Sox2 and T indicate successful differentiation into NMPs. After 48 hr of differentiation of EpiSCs into NMPs media was switched to N2B27 supplemented with 20 ng/ml Fgf2 to generate prospective paraxial mesoderm or 20 ng/ml Fgf2 +20 ng/ml BMP4 to generate prospective lateral mesoderm.

## Mouse in situ hybridisation riboprobes

Dioxigenin labelled riboprobes were used for all in situ hybridisation experiments. The probe for Id3 is described in *Jen et al. (1997)* and the probe for Id1 is described in *Gray et al. (2004)*.

Mouse grafting and embryo culture

Grafting and embryo culture was carried out as described previously (*Wymeersch et al., 2016*).

## Antibodies

- Rabbit α-Sox2 from Abcam (ab97959)
- Goat α-T from R and D systems (AF2085)
- Goat α-Meox1 (M-15) from Santa Cruz (sc-10185)
- Rabbit α-Sox1 from Cell Signalling (#4194)
- Mouse α-flag from BioM2
- Rabbit anti-Id1 clone 37–2 from Biocheck Inc qPCR

Primers used in qPCR experiments are listed below. All qPCR data represents data from three independent experiments other than *Figure 6D* which represents data from two independent clonal lines.

## Grafting of cultured mouse cells

r04-GFP EpiSCs were used for all grafts of cultured cells described here (*Huang et al., 2012*). This cell line was derived directly from post-implantation mouse embryo epiblast. The line contains constitutively expressed GFP. Prior to in vitro differentiation into NMPs r04-GFP EpiSCs were subject to fluorescent activated cell sorting to eliminate any cells that had silenced the fluorescent label. GFP positive cells were plated into EpiSC culture conditions for 48 or 72 hr prior to differentiation into NMPs. NMP differentiation was performed as described in *Gouti et al. (2014)*, and cells grafted following 48 hr of NMP differentiation. Gentle scraping using a 20–200 µl pipette tip was used to detach clumps of cells from adherent culture. These clumps were sucked into hand drawn glass pipettes, which were used to graft the cells into host embryos. Host embryos were held in place gently with forceps and the graft-containing pipette inserted into the embryo at the desired graft site. The graft was expelled as the pipette was gently drawn out of the embryo. Embryos were then imaged as a record of the graft site and transferred into culture.

| Gene name | F primer | R primer |
|---|---|---|
| T | ACTGGTCTAGCCTCGGAGTG | TTGCTCACAGACCAGAGACTG |
| Wnt3a | AATGGTCTCTCGGGAGTTTG | CTTGAGGTGCATGTGACTGG |
| Hand1 | CAAGCGGAAAAGGGAGTTG | GTGCGCCCTTTAATCCTCTT |
| Meox1 | AGACGGAGAAGAAATCATCCAG | CTGCTGCCTTCTGGCTTC |
| Kdr (Flk1) | CCCCAAATTCCATTATGACAA | CGGCTCTTTCGCTTACTGTT |
| Id1 | TCCTGCAGCATGTAATCGAC | GGTCCCGACTTCAGACTCC |
| Tcf15 | GTGTAAGGACCGGAGGACAA | GATGGCTAGATGGGTCCTTG |
| Oct4 | GTTGGAGAAGGTGGAACCAA | CTCCTTCTGCAGGGCTTTC |
| Sox2 | GTGTTTGCAAAAAGGGAAAAGT | TCTTTCTCCCAGCCCTAGTCT |

## Acknowledgements

We thank Cecilia Moens and Charles Kimmel for fish strains, Scott Houghtaling for probes, Neal Bhattacharji for zebrafish care, and Mattias Malaguti for mouse Id1-reporters and Id1-inducible cell lines. This work was supported by MRC grant Mr/K011200/1 to VW, Wellcome Trust Senior Fellowship in Basic Biomedical Science 103789/Z/14/Z to SL, Seattle Children's Research Institute Myocardial Regeneration Initiative and the NIH (R03AR065760) to LM, and Stony Brook University Startup Funds, American Heart Association grant 13SDG14360032, and NSF grant IOS1452928 to BLM.

## Additional information

### Funding

| Funder | Grant reference number | Author |
|---|---|---|
| National Science Foundation | CAREER Award IOS1452928 | Benjamin Louis Martin |
| American Heart Association | National Scientist Development Grant 13SDG14360032 | Benjamin Louis Martin |
| National Institute of Arthritis and Musculoskeletal and Skin Diseases | R03 AR065760 | Lisa Maves |
| Medical Research Council | Mr/K011200/1 | Valerie Wilson |

| Wellcome | Senior Fellowship in Basic Biomedical Science 103789/Z/14/Z | Sally Lowell |

The funders had no role in study design, data collection and interpretation, or the decision to submit the work for publication.

## Author contributions
Richard H Row, Amy Pegg, Conceptualization, Data curation, Formal analysis, Investigation, Visualization, Methodology, Writing—review and editing; Brian A Kinney, Data curation, Investigation, Methodology, Writing—review and editing; Gist H Farr 3rd, Formal analysis, Investigation, Writing—review and editing; Lisa Maves, Formal analysis, Supervision, Funding acquisition, Investigation, Writing—review and editing; Sally Lowell, Valerie Wilson, Benjamin L Martin, Conceptualization, Resources, Data curation, Formal analysis, Supervision, Funding acquisition, Investigation, Methodology, Writing—original draft, Project administration, Writing—review and editing

## Author ORCIDs
Sally Lowell ⬚ http://orcid.org/0000-0002-4018-9480
Valerie Wilson ⬚ http://orcid.org/0000-0003-4182-5159
Benjamin L Martin ⬚ http://orcid.org/0000-0001-5474-4492

## Ethics
Animal experimentation: This study was performed in accordance with and approval from the Stony Brook University Institutional Animal Use and Care Committee (IACUC) (protocol # 301537), Seattle Children's Research Institute IACUC (protocol # 14109), and the Animal Welfare and Ethical Review Panel of the MRC Centre for Regenerative Medicine and within the conditions of the Animals (Scientific Procedures) Act of 1986 (UK Home Office project license PPL60/4435).

## Decision letter and Author response
Decision letter https://doi.org/10.7554/eLife.31018.029
Author response https://doi.org/10.7554/eLife.31018.030

# Additional files

## Supplementary files
• Transparent reporting form
DOI: https://doi.org/10.7554/eLife.31018.028

## Data availability
Transgenic fish lines generated for this study will be made available through the Zebrafish International Resource Center. The raw data for the qPCR experiments is available as Figure 2—source data 1, Figure 5—source data 1 and Figure 6—source data 1.

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
