## [Decision Letter]

Thank you for submitting your article "BMP and FGF signaling interact to pattern mesoderm by controlling bHLH transcription factor activity" for consideration by *eLife*. Your article has been reviewed by three peer reviewers, and the evaluation has been overseen by a Reviewing Editor and Marianne Bronner as the Senior Editor. The following individual involved in review of your submission has agreed to reveal her identity: Anna-Katerina Hadjantonakis (Reviewer #2).

The reviewers have discussed the reviews with one another and the Reviewing Editor has drafted this decision to help you prepare a revised submission. *eLife*

Summary:

In this manuscript, Row et al. present a compelling analysis of the way in which FGF and BMP signaling intersect to pattern the vertebrate mesoderm. FGF is known to activate expression of bHLH transcription factors that are required for paraxial mesoderm development, such as Mesogenin, Myod and Myf5. The authors show that BMP promotes ventral mesoderm (vasculature) by activating the expression of Id, a known inhibitor of bHLH proteins. Strikingly, simultaneous inhibition of Mesogenin, Myod and Myf5 causes paraxial tissue to adopt a vascular fate. It is especially appealing that the authors employ a cross-species approach, combining work in zebrafish, mouse, and IPSC-derived mesodermal progenitors, to investigate the intersection between BMP/FGF signaling, bHLH proteins, and Id function during this important patterning process. Their interesting results suggest a conserved function for BMP signaling in inhibiting the development of neural tissue development and dorsal somitic mesoderm while also specifying endothelial cells through bHLH proteins, including Id acting as a BMP target gene to specify multiple ventral mesodermal cell fates. The authors try to integrate all of their results into a unified model for how distinct signaling impacts D-V (but not A-P) patterning of nascent mesodermal cell types emanating at gastrulation or in the tail bud. This model has important implications for our understanding of embryonic patterning and mesodermal differentiation. However, in its current form, the manuscript is challenging to read and hard to digest into a simple model (presented – but kind of buried – in Figure 8X). In addition, there are several open issues related to the authors' controls, experimental design, and results, and further work will be needed to strengthen the case for the authors' interpretations. Finally, some of the authors' conclusions seem to extend far beyond what is indicated by the available data, and it would be valuable for the authors to re-think precisely what they want to conclude (as mentioned in certain suggestions for revision listed below).

Essential revisions:

1) It is unclear why the authors use dorsomorphin for this study. It has a well-documented effect on VEGF signaling, thus the results cannot be interpreted simply as a loss of BMP signaling. The dorsomorphin data should be removed and replaced with experiments using DMH1 which has higher specificity and inhibits the BMP receptor.

2) The authors cannot conclude that BMP signaling is cell-autonomously required for endothelial cell fate specification as they do in the last paragraph of the subsection “BMP signaling is necessary and sufficient for endothelial specification from NMP derived mesoderm”, since endothelial cells with DN-BMPR still form and are reduced to half, not to 0 percent. If the authors want to test cell autonomy, it would be better done by transplanting BMP component null mutant cells in such an experiment, rather than using the hsDN-BMPR, which may not block all BMP signaling and therefore not block all from endothelial development. The experiment is still a good one, but it would be appropriate to soften the conclusion on cell autonomy. Additionally, cell non-autonomous effects would not be observed when measuring fli1-GFP endothelial cells, since only the donor cells contain the marker.

3) The *HS:dnfgfr* transplant analysis should be quantified, as with the *HS:caalk6* and *hs:dnbmpr* transplants. Same issue here with making conclusions about cell autonomy. Cell non-autonomous effects would not be observed because the *fli1-GFP* is only in the donor cells, and not all donor cells change fate, so not most rigorous test of cell autonomy. Clearly FGF signaling is playing a role and it is likely to be cell-autonomous, but these experiments are not definitive.

4) In the first paragraph of the subsection “FGF signaling prevents presomitic mesoderm from adopting an endothelial fate” and Supplementary Figure 2, without a quantitative assay for the number of *etv2*-expressing cells, it cannot be said that loss of FGF and Wnt signaling cause a synergistic increase in *etv2* cells. A change in morphology of the tailbud may also occur, so the red bar between *etv2* cells and the end of the tailbud is not sufficient a measure. The distance from *etv2* cells to the dorsal edge of the embryo appears smaller in Supplementary Figure 2C compared to 2D, possibly indicating a morphology change.

5) For experiments in Figure 4—figure supplement 1, these would be most convincing if done blindly (if not already done this way) by mixing the embryos in one tube, heat shocking, followed by in situ hybridization, and then genotyping. Also, in situ hybridizations are not a good assay for quantitative changes, since length of staining can strongly affect the amount of staining. Quantitative RT-PCR would be an alternative to the blind experiment suggested, which provides better controls for fixing, staining, etc.

6) Based on comparisons of Figure 8V (*HS:id3*) and 8W (*HS:caalk6*), it is concluded that Id3 does not affect AP patterning, while overexpression of BMP does. There are few controls or analysis performed to substantiate this conclusion. Is Id3 expressed widely and sufficiently at shield stage compared to *caalk6*? Caalk6 may act cell non-autonomously because it likely will induce BMP gene expression during gastrulation where ectopically expressed, which Id3 may not do. Much more analysis should be performed if the authors would like to make such a conclusion.

7) The Myogenin/Myod/Myf5 experiments clearly show their requirement for muscle development, and in their absence the cells form endothelium or blood. But that doesn't show that these myogenic genes inhibit 'ventral mesodermal fate'. Perhaps it is not useful to refer to Myogenin/Myod/Myf5 or Gata1a as dorsalizing or ventralizing the mesoderm. Rather, they are acting in specific cell fate decisions. Myogenin/Myod/Myf5 have a long history and are well known for their roles in specifying somitic mesoderm. Referring to them as dorsalizing factors is confusing. These factors are the outputs acting at more specific cell fate decisions, so it would be better to discuss them in that way. In the last paragraph of the subsection “Id genes are the essential targets mediating BMP induced ventralization of mesoderm”, the authors are trying to argue that their results show that Myogenin/Myod/Myf5 play roles other than in somite-muscle specification, which does not seem well supported by their results. On the other hand, Id genes may be acting more broadly and may therefore be appropriate to discuss in terms of ventralization/dorsalization.

8) One major question that comes to mind is the specificity of Id proteins. Do they bind to and inhibit all bHLH proteins? If so, then one can assume that they act by preventing Mesogenin et al. from activating transcription. However, if they have some specificity, it would be good to show the physical interaction, since the authors' conclusions assume that Id is sequestering mesogenin et al. Bimolecular fluorescence complementation has been used to show binding specificity between hairy-related proteins (also bHLH proteins) in live zebrafish and could be employed in a similar manner here.

9) In the Discussion, the authors ask whether Id is a morphogen. The answer is no. A morphogen diffuses from a source and specifies at least two cell fates above a ground state in a concentration-dependent manner. Given that the vertebrate embryo is not a syncytium and that Id is a transcription factor, it cannot function as a morphogen. BMPs can function as morphogens and, perhaps as an effector of Bmp signaling, Id can function to specify multiple cell fates, which seems to be what the authors are trying to say in the discussion. However, Id cannot be a morphogen and the authors should use this terminology appropriately.

10) Regarding the discussion: "Distinct molecular control of AP and DV axis formation downstream of FGF and BMP signaling": The role of BMP signaling in AP patterning is very distinct from the roles of Wnt, FGF, RA in AP patterning of the embryo and needs to be clarified. As written, it could be misleading to many readers and confusing to those more familiar with the roles of Wnt and RA that pattern AP tissues. Because tail tissues are derived predominantly from ventral regions of the gastrula, they are affected by BMP loss or gain of function. I think it is worth clarifying this, so the distinct roles of these signaling pathways is conveyed. The discussion on BMP signaling controlling AP and DV patterning of mesodermal tissue should also be discussed in terms of loss of function studies. A few studies examining AP patterning of neural tissue shows that it is unaffected in BMP loss of function mutants, related to the prior point.

11) "Our results show for the first time a molecular pathway downstream of BMP and FGF signaling that controls DV pattern without affecting AP pattern." It does not seem that this conclusion, which is also in the Abstract, can be made based on the Figure 8 results.

12) Although the Id genes are well known as BMP direct targets, it does not seem like they have been shown to be direct targets in these tissues for zebrafish. The authors should qualify their conclusions and say, for example, presumptive (or likely) direct target. A role for different levels of Id protein in patterning DV tissues is stated in the Discussion (subsection “Do ID proteins act as morphogens?”, first paragraph), although it's unclear which experiments demonstrated this. Since the Hs:Id experiments do not show this, this conclusion seems likely to be an overstatement. Later in this paragraph two mechanisms by which Id could act is discussed, which is good, but the previous statement should be removed.

13) The cross-species approach employed here is a great strength of this work. However, although this manuscript tries to propose an integrated model of signaling between zebrafish and mouse, much of the insight comes from the fish experiments, which are somehow disconnected from the mouse and pluripotent stem cell work. It is not expected that all work will be performed in parallel in all contexts (e.g. it is not necessary for all fish experiments to be recapitulated in the mouse), but it is important for the authors to integrate their coverage of the experiments performed in each system as well as they can. It will also be worthwhile for the authors to make each part of the manuscript as accessible as possible (e.g. making the zebrafish portions of the manuscript highly accessible to readers from the mouse field, and vice versa). In this regard, the authors should consider how they can revise their manuscript to address the following comments from the reviewers:

a) This study specifically identifies and characterizes a role of BMP and FGF signaling in D-V (but not A-P) patterning. While BMP acting as a ventralizing signal may be well described in zebrafish, it is not well documented in the mouse, and so many of the analogies the authors make are hard to reconcile between the two species making it even more difficult for the reader to integrate.

b) The introduction to the manuscript presents a discussion of NMPs. However, it is somewhat polarized and centered on studies in the zebrafish. It neglects to mention that NMPs were first identified through studies in mice, which were in many people's opinion spearheaded by a tour-de-force and landmark fate mapping study (Tsouanacou et al. 2009).

c) The analysis of gastrulation is only performed (or extended) to zebrafish. This is fine, although it sort of stands out.

d) The mouse grafting experiments (Figure 2A) seem a little disjointed with the rest of the study and could be better explained and integrated into the big picture of the story.

[Editors' note: further revisions were requested prior to acceptance, as described below.]

Thank you for resubmitting your work entitled "BMP and FGF signaling interact to pattern mesoderm by controlling basic helix-loop-helix transcription factor activity" for further consideration at *eLife*. Your revised article has been favorably evaluated by Marianne Bronner (Senior Editor), Deborah Yelon (Reviewing Editor), and three reviewers.

All of the reviewers agree that the manuscript has been improved through revision. The new data provided are helpful to address the prior critiques, the conclusions are well substantiated by the data, and the overall findings are quite interesting and provide a more integrated understanding of how FGF and BMP signaling pattern the vertebrate mesoderm. However, there are some remaining issues that need to be addressed before acceptance, as outlined below:

1) An important issue remains regarding whether Id and Myod/Myog/Myf5 function during gastrulation. Although the experiments (subsection “bHLH transcription factor activity functions during zebrafish gastrulation to pattern the mediolateral mesodermal axis”, first paragraph) are convincing that triple loss of *myod, myog*, and *myf5* leads to loss of skeletal muscle and an increase in endothelial cells and blood, the results do not support a role for an alteration in cell fate specification during gastrulation. Analysis was done at 24 hpf or 22-somite stage, which is a long time from gastrulation. These genes are expressed throughout this period, so it seems a leap in logic to make conclusions about gastrulation, without examining much earlier stages in development. This is a weak point and these experiments are not evidence for function in gastrulation.

The *HS:id3* experiment where heat-shock is done at shield stage provides some evidence for a role in gastrulation; however, it is not conclusive. It would have been desirable to show that the same expansion in blood was not observed with a heat-shock after gastrulation. The persistence of the transcript and Id3 protein beyond gastrula stages could lead to the expansion, rather than actual patterning during gastrulation. Indeed, Figure 8—figure supplement 1 supports long persisting Id3 protein, as an early heat-shock (shield, 75% epiboly) affects cell fates extensively along the AP axis. In addition, heat-shock at 75% epiboly (mid-gastrula) looks nearly identical to heat-shock at shield stage (early gastrula), indicating that Id3 is sufficient at late gastrula stages to inhibit somite and promote endothelial cell fates. Further, heat-shock after gastrulation at bud stage still strongly affects anterior somites (some myoD expression anteriorly but still significantly reduced compared to WT) and shows an expansion of *kdrl* (endothelial) expression along the entire AP axis, including the most anterior. Thus, it seems that the results do not support a role for patterning during gastrulation. Here too patterning was only analyzed at 24 hpf, rather than during gastrulation or at early somite stages, so concluding that patterning is affected during gastrulation should be softened here or more experiments need to be performed to rigorously test this point. This is a minor point of the manuscript, and does not affect the main, important conclusions of the paper. Perhaps the simplest solution would be for the authors to remove the conclusion that patterning is occurring during gastrulation (or at least to substantially tone down this interpretation and to discuss the appropriate caveats.)

2) The authors should also comment on where the *id1* and *id3* genes are expressed in zebrafish. ZFIN shows ubiquitous expression of *id1* during gastrulation and *id3* expressed strongly in the dorsal midline with weaker ventral levels. It seems unclear if these expression domains fit well with the authors' models.

3) The style of the text still leaves something to be desired. Both clarity and precision could be improved. Below are a few specific examples.

- In the legend of Figure 1, it should be specifically noted that panels F, G, K and L are transverse sections.

- In the legend of Figure 4, panels G and H should be specifically noted.

- In the last paragraph of the subsection “Lateral mesoderm is the default state of zebrafish tailbud derived mesoderm” and in the legend of Figure 3, it is stated that inhibition of BMP signaling failed to "rescue" MEK inhibitor mediated etv2 expansion. "rescue" is a confusing verb here. "prevent" or 'block' would be much clearer.

- The authors write, "Together, the zebrafish and mouse data show that BMP mediated lateralization of NMP derived mesoderm occurs through Id gene activation across vertebrates." A more precise phrasing would be, "The mouse and zebrafish data show that BMP mediates lateralization of NMP derived mesoderm via activation of Id expression and suggests that this mechanism acts across vertebrates."

- In the first paragraph of the subsection “bHLH transcription factor activity functions during zebrafish gastrulation to pattern the mediolateral mesodermal axis”, it should refer to the transplant analysis as 'chimeric' rather than 'mosaic', since these are chimeras (cells from two different embryos) and not mosaic embryos (which would be all cells from the same embryo but genetically different). For this analysis in Figure 8G-J, if the numbers of cells are known of each cell type, please include. Although the result is convincing in the two embryos shown, quantitation is always more rigorous.

- Supplementary figures are labelled differently than in the text, making it a bit confusing to determine which goes with what.

- Subsection “Evolutionarily conserved mediolateral patterning of NMP derived mesoderm”, first paragraph, 'medializing' should be 'lateralizing'.

- There are still many exceedingly long paragraphs that should be separated into multiple paragraphs.

---

## [Author Response]

Essential revisions:1) It is unclear why the authors use dorsomorphin for this study. It has a well-documented effect on VEGF signaling, thus the results cannot be interpreted simply as a loss of Bmp signaling. The dorsomorphin data should be removed and replaced with experiments using DMH1 which has higher specificity and inhibits the BMP receptor.

We repeated the dorsomorphin experiments using DMH1 as suggested and found that the effects on medial-lateral patterning of the NMP derived mesoderm were the same as what we originally observed using dorsomorphin and *HS:dnbmpr*. The new DMH1 data can be found in Figures 1, 3, and Figure 1—figure supplement 1.

2) The authors cannot conclude that BMP signaling is cell-autonomously required for endothelial cell fate specification as they do in the last paragraph of the subsection “BMP signaling is necessary and sufficient for endothelial specification from NMP derived mesoderm”, since endothelial cells with DN-BMPR still form and are reduced to half, not to 0 percent. If the authors want to test cell autonomy, it would be better done by transplanting BMP component null mutant cells in such an experiment, rather than using the hsDN-BMPR, which may not block all BMP signaling and therefore not block all from endothelial development. The experiment is still a good one, but it would be appropriate to soften the conclusion on cell autonomy. Additionally, cell non-autonomous effects would not be observed when measuring fli1-GFP endothelial cells, since only the donor cells contain the marker.

We agree with this comment. Our assumption is that endothelial cells that form in the presence of DN-BMPR are already committed to the endothelial fate at the time the heat-shock is applied. But, as the reviewers correctly point out, we cannot say for sure if this is the case given data in the manuscript. The question of cell autonomy is not important for the main message of the paper; therefore, as suggested, we will change the text accordingly to soften the conclusions of this experiment (subsection “BMP signaling is necessary and sufficient for endothelial specification from NMP derived mesoderm in zebrafish”, second paragraph).

3) The HS:dnfgfr transplant analysis should be quantified, as with the HS:caalk6 and hs:dnbmpr transplants. Same issue here with making conclusions about cell autonomy. Cell non-autonomous effects would not be observed because the fli1-GFP is only in the donor cells, and not all donor cells change fate, so not most rigorous test of cell autonomy. Clearly FGF signaling is playing a role and it is likely to be cell-autonomous, but these experiments are not definitive.

We performed a quantitative analysis of cell fate changes after inhibition of FGF signaling in transplanted cells. We found a significant increase in the percent of transplanted NMP derived mesodermal cells that adopt an endothelial fate after FGF inhibition. These new results are presented in Figure 3G-I. Additionally, we will modified the text to soften the conclusions about cell autonomy similar to the previous point. We indicated that while the *HS:dnfgfr* transplant results show that it functions in a cell-autonomous fashion, we cannot rule out an additional non-autonomous effect (subsection “FGF signaling prevents zebrafish presomitic mesoderm from adopting an endothelial fate”, last paragraph).

4) In the first paragraph of the subsection “FGF signaling prevents presomitic mesoderm from adopting an endothelial fate” and Supplementary Figure 2, without a quantitative assay for the number of etv2-expressing cells, it cannot be said that loss of FGF and Wnt signaling cause a synergistic increase in etv2 cells. A change in morphology of the tailbud may also occur, so the red bar between etv2 cells and the end of the tailbud is not sufficient a measure. The distance from etv2 cells to the dorsal edge of the embryo appears smaller in Supplementary Figure 2C compared to 2D, possibly indicating a morphology change.

This figure was intended to link this work with some of our prior work (Martin and Kimelman, 2012) in which we showed a role for canonical Wnt signaling during the paraxial / endothelial fate decision. Since this is the only figure that examines Wnt function and does not add to the main message of the paper, which focuses on FGF and BMP interactions, and given the reviewers concerns, we decided to remove this supplemental figure from the manuscript. We plan to integrate the Wnt data into subsequent work that will take a more in-depth approach to understanding the role of Wnt signaling in this cell fate decision.

5) For experiments in Figure 4—figure supplement 1, these would be most convincing if done blindly (if not already done this way) by mixing the embryos in one tube, heat shocking, followed by in situ hybridization, and then genotyping. Also, in situ hybridizations are not a good assay for quantitative changes, since length of staining can strongly affect the amount of staining. Quantitative RT-PCR would be an alternative to the blind experiment suggested, which provides better controls for fixing, staining, etc.

We repeated these experiments as the reviewers suggested by performing a blind analysis of mixed embryos that were PCR genotyped after in situ hybridization and image capture. Embryos were sorted based on whether they exhibited strong, medium, or weak expression of *id1* or *id3*. The embryos were subsequently PCR genotyped using primers specific for the *HS:dnbmpr* or *HS:caalk6* transgenes. In the vast majority of embryos, strong expression of either *id1* or *id3* correlated with the presence of the *HS:caalk6* transgene, weak expression correlated with the presence of the *HS:dnbmpr* transgene, and medium expression with the absence of both transgenes. These results are presented in Figure 4—figure supplement 1G and H, with numbers given in legend.

6) Based on comparisons of Figure 8V (HS:id3) and 8W (HS:caalk6), it is concluded that Id3 does not affect AP patterning, while overexpression of BMP does. There are few controls or analysis performed to substantiate this conclusion. Is Id3 expressed widely and sufficiently at shield stage compared to caalk6? Caalk6 may act cell non-autonomously because it likely will induce bmp gene expression during gastrulation where ectopically expressed, which Id3 may not do. Much more analysis should be performed if the authors would like to make such a conclusion.

The reviewers raise a valid point here. Although we feel this is potentially a very interesting result, the differences in the effect of Id3 and BMP are not central to the conclusion regarding the mediolateral mesoderm patterning mechanism presented in the manuscript, and thus we removed this conclusion and will expand upon it in future work. Furthermore, as discussed below, other comments from the reviewers (e.g. point 7 below) have helped us to realize that the broad focus on dorso-ventral patterning would be better framed in terms of a specific focus on lateral vs. medial (paraxial) mesoderm patterning. This makes it even clearer that this particular conclusion is only marginally relevant the overall message of the paper. We removed panels V and W from Figure 8 and adjusted the text accordingly to reflect these changes.

7) The Myogenin/Myod/Myf5 experiments clearly show their requirement for muscle development, and in their absence the cells form endothelium or blood. But that doesn't show that these myogenic genes inhibit 'ventral mesodermal fate'. Perhaps it is not useful to refer to Myogenin/Myod/Myf5 or Gata1a as dorsalizing or ventralizing the mesoderm. Rather, they are acting in specific cell fate decisions. Myogenin/Myod/Myf5 have a long history and are well known for their roles in specifying somitic mesoderm. Referring to them as dorsalizing factors is confusing. These factors are the outputs acting at more specific cell fate decisions, so it would be better to discuss them in that way. In the last paragraph of the subsection “Id genes are the essential targets mediating BMP induced ventralization of mesoderm”, the authors are trying to argue that their results show that Myogenin/Myod/Myf5 play roles other than in somite-muscle specification, which does not seem well supported by their results. On the other hand, Id genes may be acting more broadly and may therefore be appropriate to discuss in terms of ventralization/dorsalization.

These comments are very helpful. We made changes to the text to be specific about the roles of Msgn1/Myod/Myf5 in somite/muscle specification and eliminate mention of them acting as dorsalizing factors in the mesoderm. This also helps to clarify the parallels between the zebrafish and the mouse work raised in point 13 and improves the overall clarity of the manuscript.

8) One major question that comes to mind is the specificity of Id proteins. Do they bind to and inhibit all bHLH proteins? If so, then one can assume that they act by preventing Mesogenin et al. from activating transcription. However, if they have some specificity, it would be good to show the physical interaction, since the authors' conclusions assume that Id is sequestering mesogenin et al. Bimolecular fluorescence complementation has been used to show binding specificity between hairy-related proteins (also bHLH proteins) in live zebrafish and could be employed in a similar manner here.

We apologize that we did not explain the mechanism by which Id inhibits bHLH proteins. Id proteins do not directly bind bHLH transcription factors such as Mesogenin in order to inhibit their activity. Rather, Id proteins bind and sequester E proteins, which are essential heterodimerisation partners of bHLH transcription factors. E proteins are expressed ubiquitously so we would not expect any specificity of Id proteins with respect to particular bHLH transcription factors. We have clarified this in the revised version of our manuscript (subsection “BMP signaling lateralizes mesoderm through activation of id gene transcription in both mouse and zebrafish”, first paragraph).

9) In the Discussion, the authors ask whether Id is a morphogen. The answer is no. A morphogen diffuses from a source and specifies at least two cell fates above a ground state in a concentration-dependent manner. Given that the vertebrate embryo is not a syncytium and that Id is a transcription factor, it cannot function as a morphogen. BMPs can function as morphogens and, perhaps as an effector of BMP signaling, Id can function to specify multiple cell fates, which seems to be what the authors are trying to say in the discussion. However, Id cannot be a morphogen and the authors should use this terminology appropriately.

Agreed. The intention was to suggest that differential Id levels may lead to alternative cell fates, and thus might be the downstream effector accounting for the morphogenetic activity of BMP during mesoderm patterning. We changed the text to clarify this in the Discussion and eliminate reference of Id as potentially being a morphogen itself (Discussion subsection “Id genes are the essential targets mediating BMP induced lateralization of mesoderm”, last paragraph).

10) Regarding the discussion: "Distinct molecular control of AP and DV axis formation downstream of FGF and BMP signaling": The role of BMP signaling in AP patterning is very distinct from the roles of Wnt, FGF, RA in AP patterning of the embryo and needs to be clarified. As written, it could be misleading to many readers and confusing to those more familiar with the roles of Wnt and RA that pattern AP tissues. Because tail tissues are derived predominantly from ventral regions of the gastrula, they are affected by BMP loss or gain of function. I think it is worth clarifying this, so the distinct roles of these signaling pathways is conveyed. The discussion on BMP signaling controlling AP and DV patterning of mesodermal tissue should also be discussed in terms of loss of function studies. A few studies examining AP patterning of neural tissue shows that it is unaffected in BMP loss of function mutants, related to the prior point.

The reviewers raise several good points here. Based on this and previous points, we have modified the text to focus on mediolateral patterning of the mesoderm, and thus we have removed this section which will be the focus of an ongoing study.

11) "Our results show for the first time a molecular pathway downstream of BMP and FGF signaling that controls DV pattern without affecting AP pattern." It does not seem that this conclusion, which is also in the Abstract, can be made based on the Figure 8 results.

As mentioned above, we changed the text to reflect a role in mediolateral mesoderm patterning rather than DV vs. AP patterning.

12) Although the Id genes are well known as BMP direct targets, it does not seem like they have been shown to be direct targets in these tissues for zebrafish. The authors should qualify their conclusions and say, for example, presumptive (or likely) direct target. A role for different levels of Id protein in patterning DV tissues is stated in the Discussion (subsection “Do ID proteins act as morphogens?”, first paragraph), although it's unclear which experiments demonstrated this. Since the Hs:Id experiments do not show this, this conclusion seems likely to be an overstatement. Later in this paragraph two mechanisms by which Id could act is discussed, which is good, but the previous statement should be removed.

We agree with this and modified the text as suggested.

13) The cross-species approach employed here is a great strength of this work. However, although this manuscript tries to propose an integrated model of signaling between zebrafish and mouse, much of the insight comes from the fish experiments, which are somehow disconnected from the mouse and pluripotent stem cell work. It is not expected that all work will be performed in parallel in all contexts (e.g. it is not necessary for all fish experiments to be recapitulated in the mouse), but it is important for the authors to integrate their coverage of the experiments performed in each system as well as they can. It will also be worthwhile for the authors to make each part of the manuscript as accessible as possible (e.g. making the zebrafish portions of the manuscript highly accessible to readers from the mouse field, and vice versa). In this regard, the authors should consider how they can revise their manuscript to address the following comments from the reviewers:a) This study specifically identifies and characterizes a role of BMP and FGF signaling in D-V (but not A-P) patterning. While BMP acting as a ventralizing signal may be well described in zebrafish, it is not well documented in the mouse, and so many of the analogies the authors make are hard to reconcile between the two species making it even more difficult for the reader to integrate.

As mentioned above, we modified the text to focus on lateral vs. paraxial mesoderm patterning rather than DV patterning. BMP is well documented in both species as promoting lateral mesoderm specification and thus we feel this change will make the text more unified and easier to understand.

b) The introduction to the manuscript presents a discussion of NMPs. However, it is somewhat polarized and centered on studies in the zebrafish. It neglects to mention that NMPs were first identified through studies in mice, which were in many people's opinion spearheaded by a tour-de-force and landmark fate mapping study (Tsouanacou et al. 2009).

We will modify the Introduction to include more details about mouse NMPs, including referencing the Tzouanacou paper in addition to the current NMP reviews that were cited.

c) The analysis of gastrulation is only performed (or extended) to zebrafish. This is fine, although it sort of stands out.

Although we are interested in performing a gastrula stage analysis in mouse, it is beyond the scope of the current study.

d) The mouse grafting experiments (Figure 2A) seem a little disjointed with the rest of the study and could be better explained and integrated into the big picture of the story.

In order to address this comment, we modified the text of the manuscript to improve our explanation for the rationale behind these experiments, including making sure the explanation of the manipulations is clear. We also changed the text to make sure the results are better integrated into the big picture by discussing how they are related to the fish experiments.

[Editors' note: further revisions were requested prior to acceptance, as described below.]

All of the reviewers agree that the manuscript has been improved through revision. The new data provided are helpful to address the prior critiques, the conclusions are well substantiated by the data, and the overall findings are quite interesting and provide a more integrated understanding of how FGF and BMP signaling pattern the vertebrate mesoderm. However, there are some remaining issues that need to be addressed before acceptance, as outlined below:1) An important issue remains regarding whether Id and Myod/Myog/Myf5 function during gastrulation. Although the experiments (subsection “bHLH transcription factor activity functions during zebrafish gastrulation to pattern the mediolateral mesodermal axis”, first paragraph) are convincing that triple loss of myod, myog, and myf5 leads to loss of skeletal muscle and an increase in endothelial cells and blood, the results do not support a role for an alteration in cell fate specification during gastrulation. Analysis was done at 24 hpf or 22-somite stage, which is a long time from gastrulation. These genes are expressed throughout this period, so it seems a leap in logic to make conclusions about gastrulation, without examining much earlier stages in development. This is a weak point and these experiments are not evidence for function in gastrulation.The HS:id3 experiment where heat-shock is done at shield stage provides some evidence for a role in gastrulation; however, it is not conclusive. It would have been desirable to show that the same expansion in blood was not observed with a heat-shock after gastrulation. The persistence of the transcript and Id3 protein beyond gastrula stages could lead to the expansion, rather than actual patterning during gastrulation. Indeed, Figure 8—figure supplement 1 supports long persisting Id3 protein, as an early heat-shock (shield, 75% epiboly) affects cell fates extensively along the AP axis. In addition, heat-shock at 75% epiboly (mid-gastrula) looks nearly identical to heat-shock at shield stage (early gastrula), indicating that Id3 is sufficient at late gastrula stages to inhibit somite and promote endothelial cell fates. Further, heat-shock after gastrulation at bud stage still strongly affects anterior somites (some myoD expression anteriorly but still significantly reduced compared to WT) and shows an expansion of kdrl (endothelial) expression along the entire AP axis, including the most anterior. Thus, it seems that the results do not support a role for patterning during gastrulation. Here too patterning was only analyzed at 24 hpf, rather than during gastrulation or at early somite stages, so concluding that patterning is affected during gastrulation should be softened here or more experiments need to be performed to rigorously test this point. This is a minor point of the manuscript, and does not affect the main, important conclusions of the paper. Perhaps the simplest solution would be for the authors to remove the conclusion that patterning is occurring during gastrulation (or at least to substantially tone down this interpretation and to discuss the appropriate caveats.)

We agree with this assessment and have changed the conclusion in the Abstract and body of the Results and Discussion. We now emphasize that gastrulation stage induction of Id3 has broad patterning effects throughout the mesodermal germ layer (beyond NMP derived mesoderm), and recapitulates the early ectopic activation of BMP signaling. We have done this without making a conclusion about whether the time of action is during gastrulation or at later stages.

2) The authors should also comment on where the id1 and id3 genes are expressed in zebrafish. ZFIN shows ubiquitous expression of id1 during gastrulation and id3 expressed strongly in the dorsal midline with weaker ventral levels. It seems unclear if these expression domains fit well with the authors' models.

We no longer conclude that Id3 (or other Id genes) is acting specifically during gastrulation (see response to point 1). We also show that at later time points (Figure 4—figure supplement 1 for fish, Figure 5 for mouse) that Id1 and Id3 are expressed in the appropriate regions of the embryo to fit with our model. Thus, we have not commented on the gastrula stage expression based on the modified conclusions, as it would seem out of place.

3) The style of the text still leaves something to be desired. Both clarity and precision could be improved. Below are a few specific examples.- In the legend of Figure 1, it should be specifically noted that panels F, G, K and L are transverse sections.

This is now noted in the revised manuscript.

- In the legend of Figure 4, panels G and H should be specifically noted.

This is now noted in the revised manuscript.

- In the last paragraph of the subsection “Lateral mesoderm is the default state of zebrafish tailbud derived mesoderm” and in the legend of Figure 3, it is stated that inhibition of BMP signaling failed to "rescue" MEK inhibitor mediated etv2 expansion. "rescue" is a confusing verb here. "prevent" or 'block' would be much clearer.

We have changed the wording as suggested for better clarity.

- The authors write, "Together, the zebrafish and mouse data show that BMP mediated lateralization of NMP derived mesoderm occurs through Id gene activation across vertebrates." A more precise phrasing would be, "The mouse and zebrafish data show that BMP mediates lateralization of NMP derived mesoderm via activation of Id expression and suggests that this mechanism acts across vertebrates."

We have changed the wording as suggested for better clarity.

- In the first paragraph of the subsection “bHLH transcription factor activity functions during zebrafish gastrulation to pattern the mediolateral mesodermal axis”, it should refer to the transplant analysis as 'chimeric' rather than 'mosaic', since these are chimeras (cells from two different embryos) and not mosaic embryos (which would be all cells from the same embryo but genetically different). For this analysis in Figure 8G-J, if the numbers of cells are known of each cell type, please include. Although the result is convincing in the two embryos shown, quantitation is always more rigorous.

We have changed the wording as suggested. Regarding the quantitation, we agree that this would be ideal, but unfortunately we do not have cell numbers for these.

- Supplementary figures are labelled differently than in the text, making it a bit confusing to determine which goes with what.

We have changed the labeling of the supplementary figure files.

- Subsection “Evolutionarily conserved mediolateral patterning of NMP derived mesoderm”, first paragraph, 'medializing' should be 'lateralizing'.

Thank you for noticing this, we have changed this to lateralizing.

- There are still many exceedingly long paragraphs that should be separated into multiple paragraphs.

We have gone through the manuscript and separated the long paragraphs into shorter paragraphs.